# Decision Potential Surface: A Theoretical and Practical Approximation of LLM's Decision Boundary

## Abstract

Decision boundary, the subspace of inputs where a machine learning model assigns equal classification probabilities to two classes, is pivotal in revealing core model properties and interpreting behaviors. While analyzing the decision boundary of large language models (LLMs) has raised increasing attention recently, constructing it for mainstream LLMs remains computationally infeasible due to the enormous vocabulary-sequence sizes and the auto-regressive nature of LLMs. To address this issue, in this paper we propose *Decision Potential Surface (DPS)*, a new notion for analyzing LLM decision boundary. DPS is defined on the confidences in distinguishing different sampling sequences for each input, which naturally captures the *potential* of decision boundary. We prove that the zero-height isohypse in DPS is equivalent to the decision boundary of an LLM, with enclosed regions representing decision regions. By leveraging DPS, for the first time in the literature, we propose an approximate decision boundary construction algorithm, namely $K$-DPS, which only requires $K$-finite times of sequence sampling to approximate an LLM's decision boundary with negligible error. We theoretically derive the upper bounds for the absolute error, expected error, and the error concentration between $K$-DPS and the ideal DPS, demonstrating that such errors can be trade-off with sampling times. Our results are empirically validated by extensive experiments across various LLMs and corpora.

## 1 Introduction

With the rapid advancement and remarkable success of large language models (LLMs), understanding their underlying mechanisms and behaviors has become increasingly critical (Wang et al., 2023; Conmy et al., 2023; Elhage et al., 2021; Ameisen et al., 2025; Sharkey et al., 2025; Allen-Zhu & Li, 2023; Liang et al., 2025; 2024). A key approach to demystifying the "black box" of state-of-the-art AI models involves analyzing the *decision boundary* (Rosenblatt, 1958), a fundamental concept for elucidating the characteristics of machine learning (ML) models. For LLMs, decision boundaries provide valuable insights into critical phenomena, including reasoning (Yang et al., 2025b), in-context learning (Zhao et al., 2024), hallucination (Mayne et al., 2025), memorization (Li et al., 2025), and so on.

As a foundational concept in machine learning, the decision boundary represents a subspace of inputs where a model assigns equal probability to two distinct classification outcomes (Rosenblatt, 1958). Extensive theoretical and empirical studies (Lee & Oommen, 1997; Turner & Ghosh, 1996; Goodfellow et al., 2015; Madry et al., 2018; Gu et al., 2017) have demonstrated that the properties of decision boundaries reveal critical attributes of machine learning models, including performance, robustness, and generalization. Consequently, constructing and leveraging decision boundaries for LLMs become a powerful and promising approach to enhance *almost all downstream analyses* of their behavior and capabilities.

Unfortunately, analyzing decision boundaries of LLMs incurs significantly greater complexity than on deep neural networks (DNNs) (Karimi et al., 2020; Karimi & Tang, 2020; Li et al., 2019; Lee & Landgrebe, 1997; Mickisch et al., 2020; Yousefzadeh & O'Leary, 2019). Unlike classification tasks with a limited number of classes (Lee & Oommen, 1997; Turner & Ghosh, 1996; Goodfellow

et al., 2015; Madry et al., 2018; Gu et al., 2017), LLMs predict a single token from an expansive vocabulary, often exceeding 100,000 tokens. Moreover, their autoregressive nature (Bengio et al., 2003; Radford et al., 2018) requires iterative token predictions to generate complete sequences, which further compounds the complexity of modeling decision boundaries. For instance, a Qwen-3 model (with 8 billion parameters) (Yang et al., 2025a) supports sequences up to 32,768 tokens with a vocabulary of 151,936, resulting in approximately $10^{169,790}$ decision regions! Such an enormous scale renders trivial attempts on decision-boundary-based analysis or visualization computationally infeasible. Prior studies (Zhao et al., 2024; Yang et al., 2025b; Mayne et al., 2025; Li et al., 2025), despite their valuable contributions to their specific motivating tasks, unfortunately sidestep this critical challenge. They either simplify the problem to toy scenarios, such as binary classification (Zhao et al., 2024; Mayne et al., 2025), or use the decision boundary concept metaphorically without constructing it (Yang et al., 2025b; Li et al., 2025). Consequently, the haunting questions remain unanswered — **What constitutes an LLM's decision boundary, and is there a universal and yet efficient algorithm to construct it?**

To address these questions, we propose a principled strategy for modeling the decision boundaries of LLMs, which yields theoretical guarantees, computational tractability, and interpretability simultaneously. Inspired by the existing decision boundaries for multi-class classification, we treat generative language models as a composite multi-class classification task. As trivial solutions cannot model the complex decision boundaries for such tasks, we introduce a novel concept, namely *Decision Potential Surface (DPS)*, to facilitate decision boundary analysis. It is a landscape in which every point encodes the *competition potential* among candidate outputs, quantified by a *decision potential function (DPF)*. We theoretically demonstrate that the zero-height *isohypse* of the DPS corresponds to the decision boundary, with the enclosed regions representing decision regions.

By examining the definition of DPS, we surprisingly discover that enumerating the entire output space is unnecessary for computing the DPF. Instead, a sufficient sampling already captures the "competition potential". We therefore approximate the LLM's decision boundary with only $K$-finite ($K \ll$ realistic classification count) sequence sampling, yielding $K$-*DPS* and keeping the theoretical error within a provably small bound. We establish the error bound, expected error bound, and error concentration between the ideal DPS and $K$-DPS, demonstrating that $K$-DPS offers a favorable trade-off between approximation accuracy and computational cost. Finally, we conduct extensive experiments on open-source LLMs to evaluate the empirical performance of our method.

To the best of our knowledge, this is the first study on constructing decision boundaries for LLMs. Moreover, our proposed *decision potential surface (DPS)* framework is the first to provide a practical approximation of decision boundaries with theoretical guarantees. Our contributions are as follows:

- We formalize the definition of an LLM's decision boundary as a composite multi-class classification task and analyze its fundamental properties.
- We introduce the concepts of the *Decision Potential Function (DPF)* and *Decision Potential Surface (DPS)*. We prove that the *isohypses* of the decision potential surface represent the marginal decision boundaries of LLMs, with the zero-height isohypse equivalent to the decision boundary.
- We propose $K$-DPS, an efficient and bounded approximation of the ideal DPS that requires only finite sampling for each input. We theoretically and empirically establish the error bounds of this approximation relative to the ideal DPS and quantify the trade-off between approximation error and sampling size.

## 2 RELATED WORKS

**Decision Boundary Analysis on Machine Learning Models.** The earliest exploration of decision boundaries in neural networks dates back to the era of linear classifiers and shallow architectures. Rosenblatt (1958) introduced the first linear decision boundary for binary classification, where a hyperplane separates input samples into two classes. For shallow feedforward neural networks (FFNNs) with non-linear activations (e.g., sigmoid, ReLU), subsequent work quantified how hidden layers enable non-linear decision boundaries: Lee & Oommen (1997) proposed a feature extraction method that maps input data to a space aligned with FFNN decision boundaries, showing that boundary curvature correlates with model capacity and classification accuracy. For ensemble neural

networks, Turner & Ghosh (1996) made a pivotal contribution: they developed a theoretical framework that connects the stability of decision boundaries (relative to the optimal boundary defined by Bayes theory) to the overall error performance of the ensemble. Their work showed that linearly combining multiple well-designed, unbiased neural classifiers can reduce fluctuations in decision boundaries, which in turn lowers the extra error that goes beyond the theoretical minimum (i.e., Bayes error). A critical insight established by this finding is that the geometric characteristics of decision boundaries are closely tied to how well a model performs, which sheds light on employing decision boundary to explain the properties of neural networks.

**Decision Boundary Analysis on Neural Networks.** In recent years, researchers extended boundary analysis to convolutional neural networks (CNNs) and transformers for computer vision (CV) tasks. Goodfellow et al. (2015) revealed a key vulnerability of deep CNNs: their decision boundaries are locally linear in high-dimensional input spaces, making them susceptible to adversarial examples. Madry et al. (2018) further formalized this by proving that robust training (e.g., adversarial training) "smooths" decision boundaries, reducing local linearity and adversarial susceptibility. Similarly, Gu et al. (2017) focused on backdoor attacks in CNNs, linking them to hidden "trapdoors" in decision boundaries. Such attacks involve planting a small, specific pattern that shifts the boundary and forces misclassification for triggered inputs. Lee & Landgrebe (1997) laid the groundwork by introducing decision boundary feature extraction, highlighting the boundary's role in characterizing network behavior before deep learning. Later, Yousefzadeh & O'Leary (2019) examined trained networks' decision boundaries, analyzing how architectural elements (e.g., depth and activation functions) and training data influence boundary shape, complexity, and stability, providing insights into network task performance. Mickisch et al. (2020) conducted an empirical study on deep network boundaries across CV tasks, including image classification and object detection. Via quantitative and qualitative analysis, they explored boundary behavior near correct and misclassified samples and adversarial examples, bridging theory-practice gaps. In the same year, Karimi & Tang (2020) reviewed boundary research challenges such as high input dimensionality, complex architectures, and limited visualization tools, and opportunities, including advanced math, innovative visualization, and robustness enhancements. Karimi et al. (2020) complementarily proposed metrics like smoothness, curvature, and class separation to quantify boundaries, enabling cross-model comparisons and standardized analysis for deep learning interpretability.

**Decision Boundary Analysis on Large Language Models.** Extending decision boundary analysis from traditional neural networks (e.g., CNNs) to LLMs, researchers have begun exploring how this concept illuminates LLMs' decision-making mechanisms and limitations. Zhao et al. (2024) probed the decision boundaries of in-context learning in LLMs, shedding light on how contextual information shapes boundary formation and decision outputs. Li et al. (2025) surveyed LLMs' knowledge boundaries, laying a foundation for linking knowledge scope to decision boundaries. Mayne et al. (2025) revealed LLMs' ignorance of their own decision boundaries and the unreliability of self-generated counterfactual explanations. Yang et al. (2025b) proposed BARREL, a boundary-aware reasoning framework to enhance LLM factuality via boundary awareness.

However, existing work fails to address LLM decision boundary analysis core challenges: Zhao et al. (2024) and Mayne et al. (2025) simplify to binary classification toy scenarios, deviating from LLMs' real multi-token autoregressive prediction; Yang et al. (2025b) and Li et al. (2025) use the decision boundary concept metaphorically without concrete construction methods, unable to tackle exponential decision region complexity. This leaves a lack of universal, computationally feasible, and theoretically grounded boundary-capturing approaches, hindering LLM interpretation and optimization. Therefore, this paper aims to propose new decision boundary theory for addressing high-dimensional complexity and construction barriers, enabling accurate, efficient, and interpretable boundary modeling aligned with LLMs' characteristics.

## 3 PRELIMINARIES: DECISION BOUNDARY OF LANGUAGE MODELS

### 3.1 DECISION BOUNDARY AND ITS GEOMETRY ON CLASSIFICATION MODELS

We begin our theoretical analysis with traditional classification models and aim to extend the insights to generative language models.

Consider a neural network $f : \mathbb{R}^d \to \mathbb{R}^M$ that maps an input sample $\mathbf{x} \in \mathbb{R}^d$ to a predicted probability distribution over $M$ classes, where the set of classes can be denoted as $\mathcal{M} = \{1, 2, \ldots, M\}$. $M > 2$. Our goal is to characterize the properties of $f$ under a specific input data distribution $\mathcal{D} \subseteq \mathbb{R}^d$. Without loss of generality, we decompose $f$ into three components: *(i)* A representation module $h = f_r(\mathbf{x}) : \mathbb{R}^d \to \mathbb{R}^{d'}$ that maps the input $\mathbf{x}$ to a latent representation $h$. *(ii)* A linear classification head $z = f_{\text{cls}}(h) = W_{\text{cls}}h + b_{\text{cls}} : \mathbb{R}^{d'} \to \mathbb{R}^M$, where $W_{\text{cls}} \in \mathbb{R}^{M \times d'}$ and $b_{\text{cls}} \in \mathbb{R}^M$ are learnable parameters, projecting the representation $h$ into classification logits $z$. *(iii)* A nonlinear normalization function $P = \sigma(z) : \mathbb{R}^M \to \mathbb{R}^M$, which transforms the logits into a probability distribution $P = [p_1, p_2, \ldots, p_M]$, where $0 \le p_i \le 1$ for $i = 1, \ldots, M$ and $\sum_{i=1}^{M} p_i = 1$. The final predicted class for $\mathbf{x}$ is determined by $\arg\max_i p_i$. Then, the decision boundary of the neural network is defined as follows.

**Definition 3.1** (Decision Boundary of $f$). The decision boundary of a neural network $f$ under an input distribution $\mathcal{D}$ is the set of inputs $\mathbf{x} \in \mathcal{D}$ for which at least two classes in $\mathcal{M} = \{1, 2, \ldots, M\}$ have equal and maximal prediction probabilities. Formally, we denote this set as $\mathcal{B}_M^{(f,\mathcal{D})}$, defined by:

$$\mathcal{B}_M^{(f,\mathcal{D})} = \left\{ \mathbf{x} \in \mathcal{D} \mid \exists\, m, n \in \mathcal{M}, m \ne n, \text{ such that } p_m = p_n \text{ and } p_m \ge \max_{o \in \mathcal{M} \setminus \{m,n\}} p_o \right\}, \quad (1)$$

where $p_i = P[i] = \sigma(f_{\text{cls}}(f_r(\mathbf{x})))[i]$ is the predicted probability for class $i$.

Based on Definition 3.1, we characterize the decision boundary for multi-class classification scenarios as follows.

**Theorem 3.2** (Properties of Multi-Class Classification Boundary). *For multi-class classification ($M > 2$), the decision boundary of $f$ can be expressed as:*

$$\mathcal{B}_M^{(f,\mathcal{D})} = \bigcup_{1 \le m < n \le M} \mathcal{B}_{mn},$$

$$\mathcal{B}_{mn} = \{\mathbf{x} \mid (w_m - w_n)h + (b_m - b_n) = 0,\ z_m = z_n \ge z_o\ \forall o \ne m, n,\ h = f_r(\mathbf{x}),\ \mathbf{x} \in \mathcal{D}\}. \quad (2)$$

*where $z = W_{cls}h + b_{cls}$ is the logits, $w_m$ and $w_n$ are the $m$-th and $n$-th rows of $W_{cls}$, and $b_m, b_n$ are the corresponding entries of $b_{cls}$.*

*Geometrically, $\mathcal{B}_M$ induces a Voronoi partition of the representation space, where each class corresponds to a Voronoi cell.*

The proof of Theorem 3.2 is provided in Appendix B.1.

## 3.2 DECISION BOUNDARY FOR LARGE LANGUAGE MODELS

An LLM $f : \mathcal{V}^{N_q} \to \mathcal{V}^{N_r}$ generates a sequence of tokens $\mathbf{y} = [y_1, \ldots, y_{N_r}]$, where each token $y_t \in \mathcal{V} = \{1, 2, \ldots, V\}$ is drawn from a vocabulary of size $V$, conditioned on an input prompt $\mathbf{x} = [x_1, \ldots, x_{N_q}] \in \mathcal{V}^{N_q}$. $N_q$ and $N_r$ are sequence length of the input and the generated texts. At each generation step $t$, the LLM predicts the next token $y_t$ based on the prompt and previously generated tokens, i.e., $y_t \sim P_f(y_t|\mathbf{x}, y_1, \ldots, y_{t-1})$. This single step generation can be viewed as a multi-class classification over $\mathcal{V}$, and thus, the single-token decision boundary follows Theorem 3.2. When defining the decision boundary for the entire sequence $\mathbf{y} \in \mathcal{V}^{N_r}$, we need to firstly model the joint probability of the sequence under the autoregressive process. We derive the decision boundary of LLMs from that of multi-classification, as shown below.

**Theorem 3.3** (Decision Boundary of Language Models). *The decision boundary of an LLM $f$ under an input text distribution $\mathcal{D}' \subseteq \bigcup_{n_q=1}^{N_q} \mathcal{V}^{n_q}$ is the set of prompts $\mathbf{x} \in \mathcal{D}'$ that lead to **equal generation probabilities** for at least two distinct sequences $\mathbf{y}_v, \mathbf{y}_w \in \mathcal{V}^{N_r}$, with their probabilities being maximal. Formally, the decision boundary $\mathcal{B}_{llm}^{(f,\mathcal{D}')}$ is:*

$$\mathcal{B}_{llm}^{(f,\mathcal{D}')} = \bigcup_{\mathbf{y}_v \ne \mathbf{y}_w \in \mathcal{V}^{N_r}} \mathcal{B}_{llm,vw},$$

$$\mathcal{B}_{llm,vw} = \left\{ \mathbf{x} \in \mathcal{D}' \mid P_f(\mathbf{y}_v|\mathbf{x}) = P_f(\mathbf{y}_w|\mathbf{x}) \ge \max_{\mathbf{y}_u \in \mathcal{V}^{N_r} \setminus \{\mathbf{y}_v, \mathbf{y}_w\}} P_f(\mathbf{y}_u|\mathbf{x}) \right\}, \quad (3)$$

*where $P_f(\mathbf{y}|\mathbf{x}) = \prod_{t=1}^{N_r} P_f(y_t|\mathbf{x}, y_1, \ldots, y_{t-1})$ is the joint probability of generating sequence $\mathbf{y}$ given prompt $\mathbf{x}$.*

The proof of Theorem 3.3 is provided in Appendix B.2.

While Theorem 3.3 provides a concise and intuitive definition of the decision boundary for LLMs, analyzing or computing this boundary could be computationally impossible in practice. As analyzed in Section 1, the primary challenge stems from the large vocabulary size and the autoregressive nature of sequence generation, i.e., for a generation of length $N_r$, the total number of possible sequences is $V^{N_r}$, leading to **an exponential growth in the classification space**. Specifically, the decision boundary defined in Equation equation 3 involves comparing pairs of sequences $\mathbf{y}_v, \mathbf{y}_w \in \mathcal{V}^{N_r}$, resulting in up to $\binom{V^{N_r}}{2} \approx \frac{(V^{N_r})^2}{2}$ pairwise comparisons. This is neither computationally feasible nor interpretable on subsequent visualization.

Given such intractability of directly analyzing the decision boundary defined in Theorem 3.3, a new strategy for constructing the decision boundary of large language models is essential. Specifically, we hope that this new construction satisfies the following criteria: First, it must be ***theoretically rigorous***, meaning the construction should be equivalent to or provide a bounded approximation of the decision boundary defined in Theorem 3.3, ensuring consistency with the formal definition of the boundary separating prompts that yield different output sequences. Second, the method should be ***practical***, meaning it must be computationally efficient and feasible for implementation, enabling the modeling of decision boundaries for industrial-scale LLMs with large vocabularies and long generation lengths. Third, the method should be ***interpretable***, meaning the constructed decision boundary should explicitly capture key properties of LLMs (e.g., curvature), and provide interpretable insights into phenomena observed in LLM behavior, such as output variability or robustness.

In the next section, we will introduce an approximation procedure for the decision boundary defined in Theorem 3.3, addressing these criteria to enable practical and meaningful analysis of LLMs.

## 4 DECISION POTENTIAL SURFACE: SIMPLIFYING DECISION BOUNDARY ANALYSIS ON LARGE LANGUAGE MODELS

In this section, we introduce the *Decision Potential Surface (DPS)*, a novel concept for analyzing the decision boundaries of LLMs by representing the *decision potential* of generated sequences as a surface over the input manifold. In Section 4.1, we formally define DPS and establish its relationship with the standard decision boundary formulation in LLMs. In Section 4.2, we propose $K$-grained DPS ($K$-DPS), a practical approximation of DPS, and theoretically derive its error bounds with respect to the ideal DPS.

### 4.1 DECISION POTENTIAL SURFACE OF LANGUAGE MODELS

**Definition 4.1** (Decision Potential Surface of Language Models). Given an input text distribution $\mathbf{x} \in \mathcal{D}'$ with $\mathcal{D}' \subseteq \bigcup_{n_q=1}^{N_q} \mathcal{V}^{n_q}$ and a language model $f : \mathcal{V}^{N_q} \to \mathcal{V}^{N_r}$ that generates an output sequence $\mathbf{y} = f(\mathbf{x}) = \arg\max_{\mathbf{y}_s \in \mathcal{V}^{N_r}} P_f(\mathbf{y}_s|\mathbf{x})$, we define the *decision potential function (DPF)* $\Phi_f^\infty(\mathbf{x}) : \mathcal{D}' \to \mathbb{R}_+$ as the squared difference in log-likelihoods between the top two generated sequences under the input prompt $\mathbf{x}$, i.e.,

$$
\begin{aligned}
\Phi_f^\infty(\mathbf{x}) &= \left( \min_{\mathbf{y}_w \in \mathcal{V}^{N_r}, \mathbf{y}_w \neq \mathbf{y}_v} \left[ \max_{\mathbf{y}_v \in \mathcal{V}^{N_r}} \log P_f(\mathbf{y}_v|\mathbf{x}) - \log P_f(\mathbf{y}_w|\mathbf{x}) \right] \right)^2 \\
&= \left( \log P_f(\mathbf{y}_{1*}|\mathbf{x}) - \log P_f(\mathbf{y}_{2*}|\mathbf{x}) \right)^2,
\end{aligned}
\tag{4}
$$

where $\mathbf{y}_{1*}, \mathbf{y}_{2*} \in \mathcal{V}^{N_r}$ denote the sequences with the highest and second-highest log-likelihoods, respectively. The *decision potential surface (DPS)* is then defined as $\mathcal{S}^{(f,\mathcal{D}')} := \{\Phi_f^\infty(\mathbf{x}) \mid \mathbf{x} \in \mathcal{D}'\}$.

Intuitively, $\mathcal{S}^{(f,\mathcal{D}')}$ can be viewed as a surface representing the competitive likelihoods across all inputs, where each decision potential value $\Phi_f^\infty(\mathbf{x})$ quantifies the *confidence* in distinguishing the most likely sequence.

Then, we define *isohypses* (i.e., contour lines) on surface $\mathcal{S}^{(f,\mathcal{D}')}$ as follows:

**Definition 4.2** ($\varepsilon$-Isohypse). The $\varepsilon$-isohypse on the decision potential surface $\mathcal{S}^{(f,\mathcal{D}')}$ is the set of inputs with the same decision potential value $\varepsilon$, i.e.,

$$\mathcal{D}'_{(\varepsilon,f)} = \{\mathbf{x} \mid \mathbf{x} \in \mathcal{D}'; \Phi_f^\infty(\mathbf{x}) = \varepsilon\}. \tag{5}$$

As a degenerate case, the zero level set of $\mathcal{S}^{(f,\mathcal{D}')}$ exhibits the following property:

**Theorem 4.3** (0-Isohypse as the Decision Boundary). *The decision boundary of a language model $f(\mathbf{x})$ under $\mathcal{D}'$, as defined in Theorem 3.3, is equivalent to the 0-isohypse, i.e.,*

$$\mathcal{B}_{llm}^{(f,\mathcal{D}')} = \mathcal{D}'_{(0,f)} = \{\mathbf{x} \in \mathcal{D}' \mid \Phi_f^\infty(\mathbf{x}) = 0\}, \tag{6}$$

*where regions separated by the 0-isohypse correspond exactly to the Voronoi cells.*

We also provide the following corollary to characterize the surface structure:

**Corollary 4.4** ($\varepsilon$-Isohypse Gives $\sqrt{\varepsilon}$-nat Confidence Hierarchy). *For any $\varepsilon > 0$, the input space $\mathcal{D}'$ is partitioned into three disjoint strata:*

- *$\varepsilon$-barriers: $\mathcal{D}'_{(>\varepsilon,f)} = \{\mathbf{x}|\Phi_f^\infty(\mathbf{x}) > \varepsilon; \mathbf{x} \in \mathcal{D}'\}$, where $f(\mathbf{x})$ predicts the sequence of its region with at least $\sqrt{\varepsilon}$ nats (natural units of information) of confidence over the next most likely sequence.*

- *$\varepsilon$-well: $\mathcal{D}'_{(<\varepsilon,f)} = \{\mathbf{x}|\Phi_f^\infty(\mathbf{x}) < \varepsilon; \mathbf{x} \in \mathcal{D}'\}$, where $f(\mathbf{x})$ has low confidence, with a margin less than $\sqrt{\varepsilon}$ nats. As $\varepsilon \to 0$, this stratum converges to the 0-isohypse.*

- *$\varepsilon$-isohypse: $\mathcal{D}'_{(\varepsilon,f)} = \{\mathbf{x} \in \mathcal{D}' \mid \Phi_f^\infty(\mathbf{x}) = \varepsilon\}$, representing the contour where the confidence margin is exactly $\sqrt{\varepsilon}$ nats.*

Proofs are provided in Appendix B.3 and B.4, respectively.

Unfortunately, computing the decision boundary or visualizing the potential surface based on Definition 4.1 and Theorem 4.3 remains computationally infeasible, as evaluating $\Phi_f^\infty(\mathbf{x})$ in Equation equation 4 requires considering all possible sequences in $\mathcal{V}^{N_r}$, resulting in a computational complexity the same as before.

Fortunately, as Equation 4 depends only on the log-likelihoods of the top two sequences, we can propose an efficient approximation with a modest error, detailed in the next subsection.

### 4.2 $K$-GRAINED DECISION POTENTIAL SURFACE AND ITS PROPERTIES

We introduce $K$-grained decision potential surface for approximating $\mathcal{S}^{(f,\mathcal{D}')}$:

**Definition 4.5** ($K$-Grained Decision Potential Surface). Given $\mathbf{x} \in \mathcal{D}'$ and a language model $f(\mathbf{x})$, we define the $K$-*grained potential function* $\Phi_f^K(\mathbf{x}) : \mathcal{D}' \to \mathbb{R}_+$ as

$$\Phi_f^K(\mathbf{x}) = \left( \min_{\mathbf{y}_w \in \mathcal{Y}_K, y_w \neq y_v} \left[ \max_{\mathbf{y}_v \in \mathcal{Y}_K} \log P_f(\mathbf{y}_v|\mathbf{x}) - P_f(\mathbf{y}_w|\mathbf{x}) \right] \right)^2 \tag{7}$$
$$= \left( \log P_f(\mathbf{y}_{1*}^K|\mathbf{x}) - \log P_f(\mathbf{y}_{2*}^K|\mathbf{x}) \right)^2,$$

where $1 \ll K \ll V^{N_r}$ denotes the size of output space for each input, $\mathcal{Y}_K = \{\mathbf{y}_v \sim P_f(\cdot|\mathbf{x})|v = 1, ..., K\}$ denotes $K$ *i.i.d.* (independent and identically distributed) sampled texts, $\mathbf{y}_{1*}^K$ and $\mathbf{y}_{2*}^K$ denotes the top-2 generated texts owning the maximal generation logarithmic likelihood within $\mathcal{Y}_K$.

In this way, the computational complexity of constructing the decision boundary is reduced from $\mathcal{O}(V^{2N_r} \cdot \mathcal{D}')$ to $\mathcal{O}(K^2 \cdot \mathcal{D}')$, resulting in a substantial reduction. This naturally leads to the next question: what is the error between $\Phi_f^K(\mathbf{x})$ and $\Phi_f^\infty(\mathbf{x})$? We address this by theoretically analyzing their relationship in the following theorems.

**Theorem 4.6** (Error Bound for Estimating $\Phi_f^\infty(\mathbf{x})$ with $\Phi_f^K(\mathbf{x})$). *For a fixed input $\mathbf{x} \in \mathcal{D}'$ and a set $\mathcal{Y}_K$ of $K$ i.i.d. samples drawn from the language model's output distribution $P_f(\cdot|\mathbf{x})$, suppose the population top-2 gap satisfies $\Delta_\infty(\mathbf{x}) = \log P_f(\mathbf{y}_{1*}|\mathbf{x}) - \log P_f(\mathbf{y}_{2*}|\mathbf{x}) \leq R_K(\mathbf{x})$, where*

$R_K(\mathbf{x}) = \log P_f(\mathbf{y}_{1*}^K|\mathbf{x}) - \min_{\mathbf{y} \in \mathcal{Y}_K} \log P_f(\mathbf{y}|\mathbf{x})$ *represents the log-likelihood diameter of $\mathcal{Y}_K$.* *Then, for any $\delta \in (0, 1)$, the error between the sample-based decision potential $\Phi_f^K(\mathbf{x})$ and the true decision potential $\Phi_f^\infty(\mathbf{x})$ satisfies:*

$$|\Phi_f^K(\mathbf{x}) - \Phi_f^\infty(\mathbf{x})| \le 2R_K^2(\mathbf{x})\sqrt{\frac{\log(4/\delta)}{2K}}, \tag{8}$$

*with probability at least $1 - \delta - 2\varepsilon_{tail}$, where $\varepsilon_{tail} = \left(1 - P_f(\mathbf{y}_{1*}^K|\mathbf{x})\right)^K$.*

**Theorem 4.7** (Expected Error Bound). *Under the same conditions of Theorem 4.6, the expected error between the sample-based decision potential $\Phi_f^K(\mathbf{x})$ and the true decision potential $\Phi_f^\infty(\mathbf{x})$ is bounded as:*

$$\mathbb{E}\left[|\Phi_f^K(\mathbf{x}) - \Phi_f^\infty(\mathbf{x})|\right] \le 2R_K^2(\mathbf{x})\sqrt{\frac{2\pi}{K}} + 4R_K^2(\mathbf{x})\varepsilon_{tail}, \tag{9}$$

*where $\varepsilon_{tail} = \left(1 - P_f(\mathbf{y}_{1*}^K|\mathbf{x})\right)^K$.*

**Corollary 4.8** (Concentration Bound). *Under the same conditions as Theorem 4.6, for any $\lambda > 0$, the tail probability of the error satisfies:*

$$\Pr\left(|\Phi_f^K(\mathbf{x}) - \Phi_f^\infty(\mathbf{x})| \ge \lambda\right) \le 4\exp\left(-\frac{K\lambda^2}{2R_K^4(\mathbf{x})}\right) + 2\varepsilon_{tail}, \tag{10}$$

*where $\varepsilon_{tail} = \left(1 - P_f(\mathbf{y}_{1*}^K|\mathbf{x})\right)^K$.*

Proofs are provided in Appendix B.5, B.6 and B.7, respectively.

From the above theorems we can see that the estimation error contracts with $K$ at the familiar $1/\sqrt{K}$ rate, mirroring the decay of an empirical mean, while the residual tail probability $\varepsilon_{\text{tail}}$ is rendered exponentially negligible by the fact that the model's top-1 sentence carries almost all mass, so the chance that it is missed in $K$ *i.i.d.* draws is effectively zero even for modest $K$. Moreover, the common factor $R_K(\mathbf{x})$ is a worst-case log-likelihood diameter whose width is dictated by the most unlikely sentence that happens to be sampled, and a sharper analysis could clearly replace this global spread with a more aggressive local gap (e.g., $R'_K(\mathbf{x}) = \log P_f(\mathbf{y}_{1*}^K|\mathbf{x}) - \log P_f(\mathbf{y}_{3*}^K|\mathbf{x})$ with $\mathbf{y}_{3*}^K$ the third most likely generated sentence in $\mathcal{Y}_K$) without inflating the bound, thereby tightening the constants while preserving the same decay.

# 5 EMPIRICAL ANALYSIS

## 5.1 SETTINGS

**Datasets and Models.** We utilize both pre-training corpora and supervised fine-tuning (SFT) datasets to simulate the input data distribution for constructing decision boundaries and the decision potential surface. For the pre-training corpus, we select Wikipedia Mini (Ridder & Schilling, 2025), an unsupervised text corpus containing a condensed version of Wikipedia articles. For supervised fine-tuning, we employ Tulu-3-SFT-MIX (Lambert et al., 2025), OpenO1-SFT (Xia et al., 2025a), HH-RLHF (Ganguli et al., 2022), and Alpaca (Taori et al., 2023), all of which are widely used in academic and industrial settings. We use Llama3.2-1B (Grattafiori et al., 2024) as the backbone in experiments.

**Implementation Details.** For sampling, we use *nucleus sampling* in our $K$-DPS implementation, with the clipping probability $p$ set to 0.9. In subsequent experiments, each data point is repeated five times. The experiments are conducted on $4 \times 80$GB Nvidia Tesla H100 GPUs.

## 5.2 INFLUENCE OF SAMPLING GRAIN $K$

We first evaluate the impact of the key hyperparameter, the sampling grain $K$, on the $K$-DPS value and the absolute errors between $K$-DPS and the ideal DPS. Specifically, we set $K = 20,000$, a sufficiently large value, to simulate and approximate the ideal DPS. We then compute the $K$-DPS values by varying $K$ from 10 to 20,000 to illustrate how the decision potential value $\Phi_f^K(\cdot)$

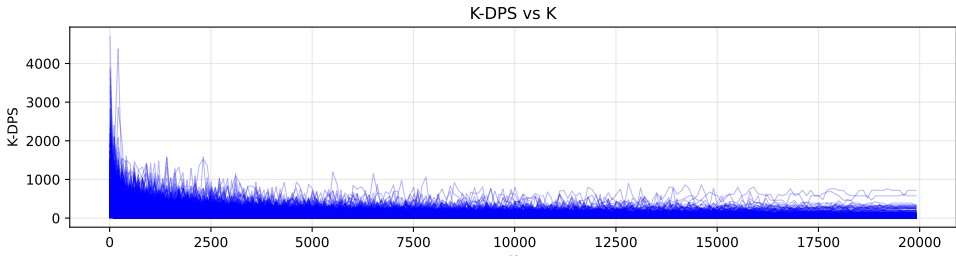

Figure 1: Effect of sampling size $K$ on the values of decision potential function, with each blue point representing the $K$-DPS value for a single input sample. Each blue line represents a trend of $K$-DPS for one input sample.

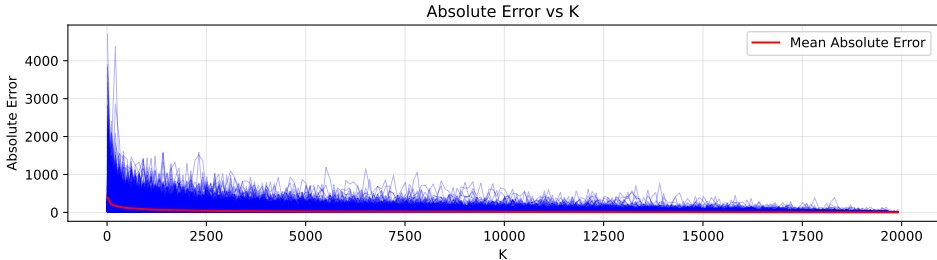

Figure 2: Effect of sampling size $K$ on the absolute error between the reference $K$-DPS (computed with $K = 20,000$) and $K$-DPS values for varying $K$. Each blue line represents a trend of absolute error across input samples.

converges to the ideal potential $\Phi_f^{\infty}(\cdot)$. Similarly, we calculate the absolute errors of $\Phi_f^K(\cdot)$ across different settings of $K$. As shown in Figure 1, the potential values rapidly converge to their true values (represented by horizontal lines at the tails), indicating that a relatively small $K$ can yield a highly accurate decision potential surface. Moreover, by examining the errors defined in Equation 8, as depicted in Figure 2, we observe that both the absolute error for individual samples and the empirically average error decrease to zero, confirming the effectiveness of $K$-DPS. Figures 1 and 2 also serve as valuable references for selecting appropriate $K$ values.

### 5.3 EMPIRICAL CONCENTRATION BIAS

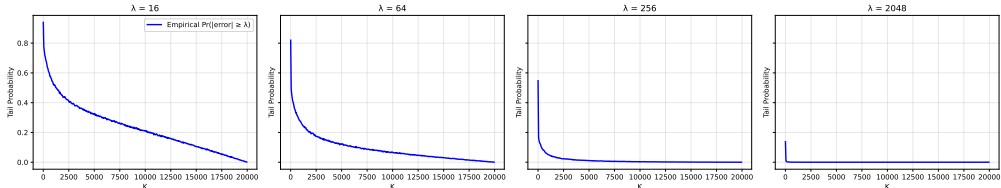

Figure 3: Empirical concentration experiments with different $\lambda$ values.

We also present an empirical study of concentration experiments, focusing on the trend of sample probabilities for inputs with a decision potential error exceeding a given fixed value $\lambda$ across various sampling sizes $K$. As shown in Figure 3, we evaluate the tail probability for $K$ values ranging from 10 to 20,000, with $\lambda$ set to 16, 64, 256, and 2048. These $\lambda$ values represent the geometric errors between the approximate and ideal DPS values. It is noteworthy to emphasize that even a $\lambda$ value of 256 is not excessively large or insignificant, as our decision potential function $\Phi_f^K(\cdot)$ is defined as the **square** of logarithmic errors, as specified in Equation 7.

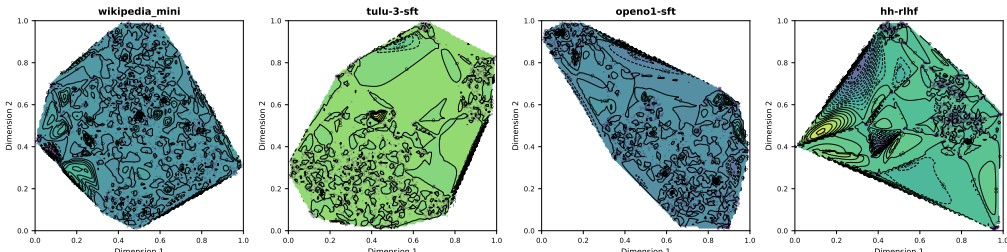

Figure 4: Contour visualization of the $K$-DPS ($K = 2,500$) for Llama-3.2-1B on four datasets. Region colors represent the decision potential values. Black lines denote isohypses, with the 0-isohypse indicating decision boundary. Cubic interpolation is applied to construct the mesh grid, with visualizations using linear and nearest interpolation shown in Figures 10 and 11.

From Figure 3, we observe that the tail probabilities exhibit an exponential decrease, indicating that the likelihood of exceeding a given error bound diminishes significantly with a linear increase in the sampling size $K$. Specifically, Figure 3 demonstrates that a sampling size of 10,000 ensures an absolute error below 64 with 90% confidence and an error below 256 with 99% probability. These results align closely with our absolute error analysis presented in Figure 2.

### 5.4 VISUALIZING DECISION POTENTIAL SURFACE

While this paper primarily focuses on the error analysis of LLMs' decision boundary construction, our proposed $K$-DPS can also be used to intuitively visualize both the decision boundary and the decision potential surface of an LLM under a given input distribution, as detailed in this section.

**Settings.** For visualization, we construct a low-dimensional representation of the original input distribution $\mathcal{D}'$, typically in two dimensions to facilitate human understanding. First, we extract the last hidden state of an input x from the LLM as the original embedding of the input point. Next, we apply UMAP with 100 neighbors and a minimum distance of 0.2 for dimensionality reduction. Finally, we normalize the reduced embeddings to the range $[0, 1]$ to construct the decision potential surface visualization. For interpolation, we evaluate nearest, linear, and cubic interpolation methods to approximate the $K$-DPS values on a mesh grid.

**Visualization Results.** As shown in Figure 4, we visualize the decision potential surface of the pretrained Llama-3.2-1B model on four corpora: Wikipedia Mini, Tulu-3-SFT, OpenO1-SFT, and HH-RLHF, using cubic interpolation. The heights of the isohypses are marked in the figures. The sampling size of $K = 2,500$. From Figure 4, it is evident that most contours are at zero height, indicating that $K$-DPS effectively captures the decision boundary of large language models. Consequently, properties of the decision boundary (i.e., the 0-isohypse), such as curvature, location, and density, can be readily analyzed to facilitate interpretation and analysis of LLMs for future studies.

However, due to limitations in the interpolation strategy, the decision potential surface may be invalid in regions with sparse input points. For instance, the top-left regions of the second and fourth subfigures show $\Phi^K$ values significantly below zero, reflecting interpolation errors due to the absence of input samples in these areas. Additional visualizations with other interpolation methods are provided in the Appendix C for reference.

## 6 CONCLUSION

In this study, we explore the construction of decision boundaries for large language models. We identify the primary challenges as stemming from the expansive vocabulary size and the exponential growth in generated token sequences. To overcome these obstacles, we propose the concept of decision potential surface to quantify the confidence of language models in their decisions. We theoretically demonstrate that the zero-height isohypse on this surface corresponds to the decision boundary, and its approximated implementation substantially reduces the computational complexity of constructing the decision boundary. Through rigorous theoretical and empirical analyses, we evaluate the errors and validate the effectiveness of the proposed method.

## REPRODUCIBILITY STATEMENT

As a theoretical study, we have clearly articulated all assumptions, definitions, theorems, and proofs in the main text and the Appendix. For the empirical results, we have provided the source code [1] to facilitate straightforward reproduction of the experimental findings. We welcome any additional suggestions to enhance the reproducibility of this work.

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

## A    LLM USAGE

It is used for error checking, proofreading, result visualization, and code optimization.

## B    PROOFS

### B.1    PROOF OF THEOREM 3.2

*Proof.* **Part I: Proof of Equation 2.**

We aim to characterize the decision boundary $\mathcal{B}_M^{(f,\mathcal{D})}$ for a neural network $f : \mathbb{R}^d \to \mathbb{R}^M$ in the multi-class classification setting ($M > 2$) under an input distribution $\mathcal{D} \subseteq \mathbb{R}^d$. The network is decomposed as $f = \sigma \circ f_{\text{cls}} \circ f_r$, where:

- $f_r : \mathbb{R}^d \to \mathbb{R}^{d_h}$ maps the input $\mathbf{x}$ to a latent representation $h = f_r(\mathbf{x})$,

- $f_{\text{cls}} : \mathbb{R}^{d_h} \to \mathbb{R}^M$ is a linear classification head, $f_{\text{cls}}(h) = W_{\text{cls}}h + b_{\text{cls}}$, with $W_{\text{cls}} \in \mathbb{R}^{M \times d_h}$, $b_{\text{cls}} \in \mathbb{R}^M$,

- $\sigma : \mathbb{R}^M \to \mathbb{R}^M$ is the softmax function, $\sigma(z)_i = \frac{e^{z_i}}{\sum_{j=1}^M e^{z_j}}$, producing probabilities $P = [p_1, p_2, \ldots, p_M]$ with $\sum_{i=1}^M p_i = 1$.

By Definition 3.1, the decision boundary $\mathcal{B}_M^{(f,\mathcal{D})}$ is the set of inputs $\mathbf{x} \in \mathcal{D}$ such that there exist at least two classes $m, n \in \mathcal{M} = \{1, 2, \ldots, M\}$, $m \neq n$, with equal and maximal probabilities:

$$p_m = p_n \geq \max_{o \in \mathcal{M} \setminus \{m,n\}} p_o. \tag{11}$$

Since $\sigma$ is the softmax function, $p_m = \sigma(z)_m = \frac{e^{z_m}}{\sum_{j=1}^M e^{z_j}}$, the condition $p_m = p_n$ implies:

$$\frac{e^{z_m}}{\sum_{j=1}^M e^{z_j}} = \frac{e^{z_n}}{\sum_{j=1}^M e^{z_j}} \implies e^{z_m} = e^{z_n} \implies z_m = z_n. \tag{12}$$

The logits are given by $z = W_{\text{cls}}h + b_{\text{cls}}$, so:

$$z_m = w_m h + b_m, \quad z_n = w_n h + b_n, \tag{13}$$

where $w_m, w_n$ are the $m$-th and $n$-th rows of $W_{\text{cls}}$, and $b_m, b_n$ are the corresponding entries of $b_{\text{cls}}$. Thus, $z_m = z_n$ implies:

$$(w_m - w_n)h + (b_m - b_n) = 0. \tag{14}$$

Additionally, for $p_m = p_n$ to be maximal, we require $p_m \geq p_o$ for all $o \neq m, n$, which implies:

$$\frac{e^{z_m}}{\sum_{j=1}^M e^{z_j}} \geq \frac{e^{z_o}}{\sum_{j=1}^M e^{z_j}} \implies e^{z_m} \geq e^{z_o} \implies z_m \geq z_o, \quad \forall o \neq m, n. \tag{15}$$

Since $z_m = z_n$, this becomes:

$$z_m = z_n \geq z_o, \quad \forall o \neq m, n. \tag{16}$$

In the representation space, this translates to:

$$(w_m - w_o)h + (b_m - b_o) \geq 0, \quad (w_n - w_o)h + (b_n - b_o) \geq 0, \quad \forall o \neq m, n. \tag{17}$$

For each pair $m, n \in \mathcal{M}$, $1 \leq m < n \leq M$, define:

$$\mathcal{B}_{mn} = \left\{ h \in \mathbb{R}^{d_h} \mid (w_m - w_n)h + (b_m - b_n) = 0, z_m = z_n \geq z_o \, \forall o \neq m, n, h = f_r(\mathbf{x}), \mathbf{x} \in \mathcal{D} \right\}. \tag{18}$$

The decision boundary is the union of all such pairwise boundaries:

$$\mathcal{B}_M^{(f,\mathcal{D})} = \bigcup_{1 \leq m < n \leq M} \mathcal{B}_{mn}. \tag{19}$$

**Part II: Voronoi Cells.**

Each $\mathcal{B}_{mn}$ is a $(d_h - 1)$-dimensional hyperplane in $\mathbb{R}^{d_h}$ defined by $(w_m - w_n)h + (b_m - b_n) = 0$, restricted to points where $z_m = z_n \geq z_o$. Geometrically, the classification region for class $i$ is:

$$\mathcal{R}_i = \left\{ h \in \mathbb{R}^{d_h} \mid w_i h + b_i > w_j h + b_j, \, \forall j \neq i, \, h = f_r(\mathbf{x}), \, \mathbf{x} \in \mathcal{D} \right\}. \tag{20}$$

These regions are convex polytopes, as they are defined by the intersection of half-spaces $(w_i - w_j)h + (b_i - b_j) > 0$. The boundaries between $\mathcal{R}_m$ and $\mathcal{R}_n$ occur where $(w_m - w_n)h + (b_m - b_n) = 0$ and $z_m = z_n \geq z_o$, forming $\mathcal{B}_{mn}$. The collection $\{\mathcal{R}_i\}_{i=1}^{M}$ partitions the representation space, and the hyperplanes $\mathcal{B}_{mn}$ form the boundaries of a Voronoi-like partition, where each $\mathcal{R}_i$ is a Voronoi cell corresponding to class $i$.

This completes the proof. $\qquad\square$

### B.2 PROOF OF THEOREM 3.3

*Proof.* We aim to characterize the decision boundary $\mathcal{B}_{llm}^{(f, \mathcal{D}')}$ of an LLM $f : \mathcal{V}^{N_q} \to \mathcal{V}^{N_r}$ under an input text distribution $\mathcal{D}' \subseteq \bigcup_{n_q=1}^{N_q} \mathcal{V}^{n_q}$. The LLM generates a sequence $\mathbf{y} = [y_1, \ldots, y_{N_r}] \in \mathcal{V}^{N_r}$, where $\mathcal{V} = \{1, 2, \ldots, V\}$ is the vocabulary, conditioned on a prompt $\mathbf{x} \in \mathcal{D}'$. The joint probability of generating $\mathbf{y}$ is:

$$P_f(\mathbf{y}|\mathbf{x}) = \prod_{t=1}^{N_r} P_f(y_t|\mathbf{x}, y_1, \ldots, y_{t-1}),$$

where $P_f(y_t|\mathbf{x}, y_1, \ldots, y_{t-1})$ is the probability of predicting token $y_t$ at step $t$, modeled as a multi-class classification over $\mathcal{V}$.

Based on $\mathbf{y}^* = \arg\max_{\mathbf{y} \in \mathcal{V}^{N_r}} P_f(\mathbf{y}|\mathbf{x})$ and Definition 3.1, the decision boundary $\mathcal{B}_{llm}^{(f, \mathcal{D}')}$ is the set of prompts $\mathbf{x} \in \mathcal{D}'$ where at least two distinct sequences $\mathbf{y}_v, \mathbf{y}_w \in \mathcal{V}^{N_r}$ have equal and maximal joint probabilities:

$$P_f(\mathbf{y}_v|\mathbf{x}) = P_f(\mathbf{y}_w|\mathbf{x}) \geq \max_{\mathbf{y}_u \in \mathcal{V}^{N_r} \setminus \{\mathbf{y}_v, \mathbf{y}_w\}} P_f(\mathbf{y}_u|\mathbf{x}).$$

For each pair of distinct sequences $\mathbf{y}_v, \mathbf{y}_w \in \mathcal{V}^{N_r}$, define:

$$\mathcal{B}_{llm,vw} = \left\{ \mathbf{x} \in \mathcal{D}' \mid P_f(\mathbf{y}_v|\mathbf{x}) = P_f(\mathbf{y}_w|\mathbf{x}) \geq \max_{\mathbf{y}_u \in \mathcal{V}^{N_r} \setminus \{\mathbf{y}_v, \mathbf{y}_w\}} P_f(\mathbf{y}_u|\mathbf{x}) \right\}.$$

The decision boundary is the union over all such pairs:

$$\mathcal{B}_{llm}^{(f, \mathcal{D}')} = \bigcup_{\mathbf{y}_v \neq \mathbf{y}_w \in \mathcal{V}^{N_r}} \mathcal{B}_{llm,vw}.$$

To show this, consider the autoregressive process. For a prompt $\mathbf{x}$, the probability $P_f(\mathbf{y}|\mathbf{x})$ depends on the token probabilities at each step. Obviously, the predicted sequence $\mathbf{y}^*$ maximizes $P_f(\mathbf{y}|\mathbf{x})$. The decision boundary occurs when two sequences $\mathbf{y}_v$ and $\mathbf{y}_w$ have equal probabilities, and no other sequence has a higher probability. This implies:

$$P_f(\mathbf{y}_v|\mathbf{x}) = \prod_{t=1}^{N_r} P_f(y_{v,t}|\mathbf{x}, y_{v,1}, \ldots, y_{v,t-1}) = \prod_{t=1}^{N_r} P_f(y_{w,t}|\mathbf{x}, y_{w,1}, \ldots, y_{w,t-1}) = P_f(\mathbf{y}_w|\mathbf{x}),$$

and for all $\mathbf{y}_u \neq \mathbf{y}_v, \mathbf{y}_w$:

$$P_f(\mathbf{y}_v|\mathbf{x}) \geq P_f(\mathbf{y}_u|\mathbf{x}).$$

Since each token prediction is a multi-class classification (as in Theorem 3.2), the boundary for a single token $y_t$ is defined by equal probabilities for the top tokens. For the full sequence, the boundary $\mathcal{B}_{llm,vw}$ corresponds to prompts $\mathbf{x}$ where the joint probabilities align, which may occur when the log-probabilities differ at some steps but sum to the same value. The maximality condition ensures that $\mathbf{y}_v$ and $\mathbf{y}_w$ are the top sequences.

This completes the proof. $\qquad\square$

## B.3 Proof of Theorem 4.3

*Proof.* We aim to prove that the decision boundary $\mathcal{B}_{llm}^{(f,\mathcal{D}')}$ defined in Theorem 3.3 is equivalent to the 0-isohypse $\mathcal{D}'_{(0,f)}$ on the decision potential surface $\mathcal{S}^{(f,\mathcal{D}')}$, and that the regions separated by this boundary correspond exactly to the Voronoi cells in the token-combined classification definition.

Recall from Theorem 3.3 that the decision boundary is

$$\mathcal{B}_{llm}^{(f,\mathcal{D}')} = \bigcup_{\mathbf{y}_v \neq \mathbf{y}_w \in \mathcal{V}^{N_r}} \mathcal{B}_{llm,vw}, \tag{21}$$

where

$$\mathcal{B}_{llm,vw} = \left\{ \mathbf{x} \in \mathcal{D}' \mid P_f(\mathbf{y}_v|\mathbf{x}) = P_f(\mathbf{y}_w|\mathbf{x}) \geq \max_{\mathbf{y}_u \in \mathcal{V}^{N_r} \setminus \{\mathbf{y}_v, \mathbf{y}_w\}} P_f(\mathbf{y}_u|\mathbf{x}) \right\}. \tag{22}$$

This boundary consists of prompts $\mathbf{x}$ where at least two distinct sequences $\mathbf{y}_v$ and $\mathbf{y}_w$ have equal and maximal joint probabilities, leading to ambiguity in the predicted output sequence.

From Definition 4.1, the decision potential function is

$$\Phi_f^\infty(\mathbf{x}) = \left( \log P_f(\mathbf{y}_1|\mathbf{x}) - \log P_f(\mathbf{y}_2|\mathbf{x}) \right)^2, \tag{23}$$

where $\mathbf{y}_1, \mathbf{y}_2 \in \mathcal{V}^{N_r}$ are the sequences with the highest and second-highest log-likelihoods, respectively. The 0-isohypse is defined as

$$\mathcal{D}'_{(0,f)} = \left\{ \mathbf{x} \in \mathcal{D}' \mid \Phi_f^\infty(\mathbf{x}) = 0 \right\}. \tag{24}$$

By definition, $\Phi_f^\infty(\mathbf{x}) = 0$ if and only if $\log P_f(\mathbf{y}_1|\mathbf{x}) = \log P_f(\mathbf{y}_2|\mathbf{x})$, which implies $P_f(\mathbf{y}_1|\mathbf{x}) = P_f(\mathbf{y}_2|\mathbf{x})$. Since $\mathbf{y}_1$ and $\mathbf{y}_2$ are the top two sequences by log-likelihood, this equality ensures that

$$P_f(\mathbf{y}_1|\mathbf{x}) = P_f(\mathbf{y}_2|\mathbf{x}) \geq P_f(\mathbf{y}_u|\mathbf{x}), \quad \forall \mathbf{y}_u \neq \mathbf{y}_1, \mathbf{y}_2, \tag{25}$$

satisfying the maximality condition in Theorem 3.3. Thus, $\mathbf{x} \in \mathcal{D}'_{(0,f)}$ if and only if $\mathbf{x} \in \mathcal{B}_{llm}^{(f,\mathcal{D}')}$, establishing the set equivalence

$$\mathcal{B}_{llm}^{(f,\mathcal{D}')} = \mathcal{D}'_{(0,f)}. \tag{26}$$

Geometrically, the regions separated by the 0-isohypse are the connected components of $\mathcal{D}' \setminus \mathcal{D}'_{(0,f)}$, where each region corresponds to prompts for which a unique sequence $\mathbf{y}_i$ has the highest log-likelihood ($\Phi_f^\infty(\mathbf{x}) > 0$). These regions are exactly the Voronoi cells in the sequence-level classification framework of Theorem 3.3, as each cell consists of prompts yielding the same maximal sequence. The 0-isohypse forms the boundaries between these cells, partitioning the prompt space $\mathcal{D}'$ into regions of unambiguous predictions.

This completes the proof. $\qquad\square$

## B.4 Proof of Corollary 4.4

*Proof.* We aim to show that for any $\varepsilon > 0$, the input space $\mathcal{D}'$ is partitioned into three disjoint strata based on the value of the decision potential function $\Phi_f^\infty(\mathbf{x})$:

$$\mathcal{D}' = \mathcal{D}'_{(>\varepsilon,f)} \sqcup \mathcal{D}'_{(<\varepsilon,f)} \sqcup \mathcal{D}'_{(\varepsilon,f)}, \tag{27}$$

where $\sqcup$ denotes disjoint union.

From Definition 4.2, the $\varepsilon$-isohypse is

$$\mathcal{D}'_{(\varepsilon,f)} = \left\{ \mathbf{x} \in \mathcal{D}' \mid \Phi_f^\infty(\mathbf{x}) = \varepsilon \right\}, \tag{28}$$

and the other strata are defined as

$$\mathcal{D}'_{(>\varepsilon,f)} = \left\{ \mathbf{x} \in \mathcal{D}' \mid \Phi_f^\infty(\mathbf{x}) > \varepsilon \right\}, \quad \mathcal{D}'_{(<\varepsilon,f)} = \left\{ \mathbf{x} \in \mathcal{D}' \mid \Phi_f^\infty(\mathbf{x}) < \varepsilon \right\}. \tag{29}$$

Since $\Phi_f^\infty : \mathcal{D}' \to \mathbb{R}_+$ is a continuous function (assuming log-likelihoods are continuous in the prompt space), these sets are disjoint and their union covers $\mathcal{D}'$.

- *$\varepsilon$-confident regions*: For $\mathbf{x} \in \mathcal{D}'_{(>\varepsilon,f)}$, $\Phi_f^\infty(\mathbf{x}) > \varepsilon$, so

$$|\log P_f(\mathbf{y}_1|\mathbf{x}) - \log P_f(\mathbf{y}_2|\mathbf{x})| > \sqrt{\varepsilon}. \tag{30}$$

Since $\mathbf{y}_1$ has the highest log-likelihood, $\log P_f(\mathbf{y}_1|\mathbf{x}) - \log P_f(\mathbf{y}_2|\mathbf{x}) > \sqrt{\varepsilon}$, meaning the model predicts $\mathbf{y}_1$ with at least $\sqrt{\varepsilon}$ nats (natural units of information) of confidence over the next most likely sequence $\mathbf{y}_2$.

- *$\varepsilon$-uncertain regions*: For $\mathbf{x} \in \mathcal{D}'_{(<\varepsilon,f)}$, $\Phi_f^\infty(\mathbf{x}) < \varepsilon$, so

$$|\log P_f(\mathbf{y}_1|\mathbf{x}) - \log P_f(\mathbf{y}_2|\mathbf{x})| < \sqrt{\varepsilon}. \tag{31}$$

Here, the model has low confidence, with a margin less than $\sqrt{\varepsilon}$ nats between the top two sequences. As $\varepsilon \to 0$, $\Phi_f^\infty(\mathbf{x}) \to 0$, so $\mathcal{D}'_{(<\varepsilon,f)}$ converges to the 0-isohypse $\mathcal{D}'_{(0,f)}$, where the margin is zero.

- *$\varepsilon$-isohypse*: For $\mathbf{x} \in \mathcal{D}'_{(\varepsilon,f)}$, $\Phi_f^\infty(\mathbf{x}) = \varepsilon$, so the confidence margin is exactly $\sqrt{\varepsilon}$ nats, forming the contour that separates confident and uncertain regions.

The disjointness of the strata follows from the strict inequalities and equality defining them, and their union covers $\mathcal{D}'$ since $\Phi_f^\infty(\mathbf{x}) \geq 0$ for all $\mathbf{x} \in \mathcal{D}'$.

This completes the proof. $\qquad\square$

### B.5 PROOF OF THEOREM 4.6

*Proof.* Let $\mathbf{x} \in \mathcal{D}'$ be a fixed input, and let $\mathcal{Y}_K = \{\mathbf{y}_1, \mathbf{y}_2, \ldots, \mathbf{y}_K\}$ be a set of $K$ *i.i.d.* samples drawn from the language model's output distribution $P_f(\cdot|\mathbf{x})$. The decision potential function is:

$$\Phi_f^\infty(\mathbf{x}) = \left(\log P_f(\mathbf{y}_{1*}|\mathbf{x}) - \log P_f(\mathbf{y}_{2*}|\mathbf{x})\right)^2, \tag{32}$$

where $\mathbf{y}_{1*}$ and $\mathbf{y}_{2*}$ are the top-2 generated texts with the highest logarithmic likelihoods over the entire output space $\mathcal{V}^{N_r}$, and

$$\Phi_f^K(\mathbf{x}) = \left(\log P_f(\mathbf{y}_{1*}^K|\mathbf{x}) - \log P_f(\mathbf{y}_{2*}^K|\mathbf{x})\right)^2, \tag{33}$$

where $\mathbf{y}_{1*}^K$ and $\mathbf{y}_{2*}^K$ are the top-2 generated texts within $\mathcal{Y}_K$. We aim to bound the error $|\Phi_f^K(\mathbf{x}) - \Phi_f^\infty(\mathbf{x})|$ with probability at least $1 - \delta - 2\varepsilon_{\text{tail}}$ for $\delta \in (0,1)$.

***Step 1*: Preliminary.** Define:

$$\begin{aligned}
\Delta_\infty(\mathbf{x}) &= \log P_f(\mathbf{y}_{1*}|\mathbf{x}) - \log P_f(\mathbf{y}_{2*}|\mathbf{x}), \\
\Delta_K(\mathbf{x}) &= \log P_f(\mathbf{y}_{1*}^K|\mathbf{x}) - \log P_f(\mathbf{y}_{2*}^K|\mathbf{x}).
\end{aligned} \tag{34}$$

Thus, $\Phi_f^\infty(\mathbf{x}) = (\Delta_\infty(\mathbf{x}))^2$ and $\Phi_f^K(\mathbf{x}) = (\Delta_K(\mathbf{x}))^2$. The error can be expressed as:

$$\begin{aligned}
|\Phi_f^K(\mathbf{x}) - \Phi_f^\infty(\mathbf{x})| &= |(\Delta_K(\mathbf{x}))^2 - (\Delta_\infty(\mathbf{x}))^2| \\
&= |\Delta_K(\mathbf{x}) - \Delta_\infty(\mathbf{x})| \cdot |\Delta_K(\mathbf{x}) + \Delta_\infty(\mathbf{x})|.
\end{aligned} \tag{35}$$

Since $\mathcal{Y}_K$ is a finite, $\mathbf{y}_{1*}^K$ and $\mathbf{y}_{2*}^K$ are the top-2 outputs in $\mathcal{Y}_K$, which may not include $\mathbf{y}_{1*}$ or $\mathbf{y}_{2*}$. Define:

$$R_K(\mathbf{x}) = \log P_f(\mathbf{y}_{1*}^K|\mathbf{x}) - \min_{\mathbf{y} \in \mathcal{Y}_K} \log P_f(\mathbf{y}|\mathbf{x}), \tag{36}$$

which represents the *diameter* of log-likelihoods in $\mathcal{Y}_K$.

**Lemma B.1** ($\Pr(\mathbf{y}_{1*} \notin \mathcal{Y}_K) \leq \varepsilon_{\text{tail}}$). *Define the tail probability $\varepsilon_{\text{tail}}$ as: $\varepsilon_{\text{tail}} = \left(1 - P_f(\mathbf{y}_{1*}^K|\mathbf{x})\right)^K$. Then, we have $\Pr(\mathbf{y}_{1*} \notin \mathcal{Y}_K) \leq \varepsilon_{\text{tail}}$,*

*A short proof of Lemma B.1:* As $\mathbf{y}_k \in \mathcal{Y}_K$ are *i.i.d.*, we know that with $K$-time sampling of $\mathbf{y} \sim P_f(\cdot|\mathbf{x})$ the probability that we cannot obtain $y_{1*}$ obeys a geometric distribution, i.e.,

$$\Pr(\mathbf{y}_{1*} \notin \mathcal{Y}_k) = (1 - P_f(\mathbf{y}_{1*}|\mathbf{x}))^K. \tag{37}$$

As $P_f(\mathbf{y}_{1*}|\mathbf{x}) \geq P_f(\mathbf{y}_{1*}^K|\mathbf{x})$, then we have

$$\Pr(\mathbf{y}_{1*} \notin \mathcal{Y}_K) = (1 - P_f(\mathbf{y}_{1*}|\mathbf{x}))^K \leq (1 - P_f(\mathbf{y}_{1*}^K|\mathbf{x}))^K = \varepsilon_{\text{tail}}, \tag{38}$$

which ends the proof.

Based on Lemma B.1, we know that $\varepsilon_{\text{tail}}$ bounds the probability that the true top output $\mathbf{y}_{1*}$ is not included in $\mathcal{Y}_K$.

***Step 2***: **Bounding $|\Delta_K(\mathbf{x}) - \Delta_\infty(\mathbf{x})|$.**

Since $\Delta_K(\mathbf{x})$ is computed over a random sample, we consider using *concentration inequalities* to bound the deviation $|\Delta_K(\mathbf{x}) - \Delta_\infty(\mathbf{x})|$. The log-likelihoods $\log P_f(\mathbf{y}_k|\mathbf{x})$ for $\mathbf{y}_k \in \mathcal{Y}_K$ are *i.i.d.*, and they are bounded within the diameter $R_K(\mathbf{x})$. By *Hoeffding's inequality*, the deviation of the sample maximum log-likelihood from its expected maximum is bounded. Specifically, for the top-1 log-likelihood $\forall\, t > 0$, we have:

$$\Pr\left(\left|\log P_f(\mathbf{y}_{1*}^K|\mathbf{x}) - \log P_f(\mathbf{y}_{1*}|\mathbf{x})\right| > t\right) \leq 2\exp\left(-\frac{2Kt^2}{R_K^2(\mathbf{x})}\right). \tag{39}$$

Similarly, for the second-highest log-likelihood, a similar bound applies.

Combining these, we have:

$$|\Delta_K(\mathbf{x}) - \Delta_\infty(\mathbf{x})| = \left|\left(\log P_f(\mathbf{y}_{1*}^K|\mathbf{x}) - \log P_f(\mathbf{y}_{2*}^K|\mathbf{x})\right) - \left(\log P_f(\mathbf{y}_{1*}|\mathbf{x}) - \log P_f(\mathbf{y}_{2*}|\mathbf{x})\right)\right|$$
$$= \left|\left(\log P_f(\mathbf{y}_{1*}^K|\mathbf{x}) - \log P_f(\mathbf{y}_{1*}|\mathbf{x})\right) - \left(\log P_f(\mathbf{y}_{2*}^K|\mathbf{x}) - \log P_f(\mathbf{y}_{2*}|\mathbf{x})\right)\right|. \tag{40}$$

Based on the triangle inequality $|a - b| \leq |a| + |b|$ when $a, b \in \mathbb{R}$, we know that

$$|\Delta_K(\mathbf{x}) - \Delta_\infty(\mathbf{x})| = \left|\left(\log P_f(\mathbf{y}_{1*}^K|\mathbf{x}) - \log P_f(\mathbf{y}_{1*}|\mathbf{x})\right) - \left(\log P_f(\mathbf{y}_{2*}^K|\mathbf{x}) - \log P_f(\mathbf{y}_{2*}|\mathbf{x})\right)\right|$$
$$\leq |\log P_f(\mathbf{y}_{1*}^K|\mathbf{x}) - \log P_f(\mathbf{y}_{1*}|\mathbf{x})| + |\log P_f(\mathbf{y}_{2*}^K|\mathbf{x}) - \log P_f(\mathbf{y}_{2*}|\mathbf{x})|. \tag{41}$$

To bound $|\Delta_K(\mathbf{x}) - \Delta_\infty(\mathbf{x})|$, we aim to find the maximal probability for the event $|\Delta_K(\mathbf{x}) - \Delta_\infty(\mathbf{x})| < t'$ with $t' > 0$. Without losing generality, we set $t' = 2t$, where the objective can be reformated as:

$$\Pr(|\Delta_K(\mathbf{x}) - \Delta_\infty(\mathbf{x})| < t')$$
$$= 1 - \Pr(|\Delta_K(\mathbf{x}) - \Delta_\infty(\mathbf{x})| >= t'), \tag{42}$$

where

$$\Pr(|\Delta_K(\mathbf{x}) - \Delta_\infty(\mathbf{x})| >= t')$$
$$= \Pr(|\log P_f(\mathbf{y}_{1*}^K|\mathbf{x}) - \log P_f(\mathbf{y}_{1*}|\mathbf{x})| \geq t \text{ or } |\log P_f(\mathbf{y}_{2*}^K|\mathbf{x}) - \log P_f(\mathbf{y}_{2*}|\mathbf{x})| \geq t))$$
$$\leq \Pr(|\log P_f(\mathbf{y}_{1*}^K|\mathbf{x}) - \log P_f(\mathbf{y}_{1*}|\mathbf{x})| \geq t) + \Pr(|\log P_f(\mathbf{y}_{3*}^K|\mathbf{x}) - \log P_f(\mathbf{y}_{2*}|\mathbf{x})| \geq t)$$
$$\leq 2\exp\left(-\frac{2Kt^2}{R_K^2(\mathbf{x})}\right) + 2\exp\left(-\frac{2Kt^2}{R_K^2(\mathbf{x})}\right)$$
$$= 4\exp\left(-\frac{2Kt^2}{R_K^2(\mathbf{x})}\right). \tag{43}$$

So we have

$$\Pr(|\Delta_K(\mathbf{x}) - \Delta_\infty(\mathbf{x})| < t')$$
$$= 1 - \Pr(|\Delta_K(\mathbf{x}) - \Delta_\infty(\mathbf{x})| >= t')$$
$$\geq 1 - 4\exp\left(-\frac{2Kt^2}{R_K^2(\mathbf{x})}\right). \tag{44}$$

Suppose we have at least $1 - \delta$ probability to support this event stands, we have

$$1 - 4\exp\left(-\frac{2Kt^2}{R_K^2(\mathbf{x})}\right) \geq 1 - \delta$$

$$\Leftrightarrow 4\exp\left(-\frac{2Kt^2}{R_K^2(\mathbf{x})}\right) \leq \delta$$

$$\Leftrightarrow \exp\left(-\frac{2Kt^2}{R_K^2(\mathbf{x})}\right) \leq \frac{\delta}{4}$$

$$\Leftrightarrow -\frac{2Kt^2}{R_K^2(\mathbf{x})} \leq \log\frac{\delta}{4}$$

$$\Leftrightarrow \frac{2Kt^2}{R_K^2(\mathbf{x})} \geq -\log\frac{\delta}{4} \tag{45}$$

$$\Leftrightarrow \frac{2Kt^2}{R_K^2(\mathbf{x})} \geq \log\frac{4}{\delta}$$

$$\Leftrightarrow t^2 \geq \frac{R_K^2(\mathbf{x})}{2K}\log\frac{4}{\delta}$$

$$\Leftrightarrow t \geq |R_K(\mathbf{x})\sqrt{\frac{log(4/\delta)}{2K}}|$$

$$\Leftrightarrow t \geq R_K(\mathbf{x})\sqrt{\frac{log(4/\delta)}{2K}}.$$

In other words, $\forall\, t > 0$ we bound $\Pr(|\Delta_K(\mathbf{x}) - \Delta_\infty(\mathbf{x})| < t)$ with probability at least $1 - \delta$ when:

$$t = R_K(\mathbf{x})\sqrt{\frac{\log(4/\delta)}{2K}}. \tag{46}$$

*Step 3*: **Bounding $|\Delta_K(\mathbf{x}) + \Delta_\infty(\mathbf{x})|$.**

**Assumption B.2** (Bounded Population Gap). There exists a constant $M > 0$ such that for any $\mathbf{x}$, the population top-2 gap satisfies:

$$\Delta_\infty(\mathbf{x}) = \log P_f(\mathbf{y}_{1*}|\mathbf{x}) - \log P_f(\mathbf{y}_{2*}|\mathbf{x}) \leq M \tag{47}$$

Then we assume that

$$M \leq R_K(\mathbf{x}) \tag{48}$$

when $K \gg 1$.

This assumption is reasonable as most practical language models do not have extremely large differences between top-2 probabilities, and the probability differences between top-2 would be much smaller than the range of between the top-1 and the sample with the minimal probaility in the sampling set. Now we can obtain that:

$$|\Delta_K(\mathbf{x}) + \Delta_\infty(\mathbf{x})| \leq |\Delta_K(\mathbf{x})| + |\Delta_\infty(\mathbf{x})| \leq 2 \cdot R_K(\mathbf{x}). \tag{49}$$

*Step 4*: **Final bound.**

Define the events

$$A = \{\mathbf{y}_{1*} \in \mathcal{Y}_K \text{ and } \mathbf{y}_{2*} \in \mathcal{Y}_K\}, \qquad B = \{\mathbf{y}_{1*} \notin \mathcal{Y}_K \text{ or } \mathbf{y}_{2*} \notin \mathcal{Y}_K\}. \tag{50}$$

Lemma B.1 and a union bound give

$$\Pr(B) \leq 2\varepsilon_{\text{tail}}. \tag{51}$$

- On event $A$ we have $\mathbf{y}_{1*}^K = \mathbf{y}_{1*}$ and $\mathbf{y}_{2*}^K = \mathbf{y}_{2*}$, hence

$$\Phi_f^K(\mathbf{x}) = \Phi_f^\infty(\mathbf{x}) \implies |\Phi_f^K(\mathbf{x}) - \Phi_f^\infty(\mathbf{x})| = 0. \tag{52}$$

- On event $B$ we use the worst-case gap

$$
\begin{aligned}
&|\Phi_f^K(\mathbf{x}) - \Phi_f^\infty(\mathbf{x})| \\
&\leq |\Delta_K(\mathbf{x}) - \Delta_\infty(\mathbf{x})| \cdot |\Delta_K(\mathbf{x}) + \Delta_\infty(\mathbf{x})| \\
&\leq R_K(\mathbf{x})\sqrt{\frac{\log(4/\delta)}{2K}} \cdot 2R_K(x) \\
&= 2R_K^2(\mathbf{x})\sqrt{\frac{\log(4/\delta)}{2K}}.
\end{aligned}
\tag{53}
$$

This completes the proof. $\qquad\square$

### B.6 PROOF OF THEOREM 4.7

*Proof.* We aim to bound the expected error $\mathbb{E}[|\Phi_f^K(\mathbf{x}) - \Phi_f^\infty(\mathbf{x})|]$ for a fixed input $\mathbf{x} \in \mathcal{D}'$ and a set $\mathcal{Y}_K = \{\mathbf{y}_1, \mathbf{y}_2, \ldots, \mathbf{y}_K\}$ of $K$ *i.i.d.* samples drawn from the language model's output distribution $P_f(\cdot|\mathbf{x})$. Recall that:

$$
\Phi_f^\infty(\mathbf{x}) = \left(\log P_f(\mathbf{y}_{1*}|\mathbf{x}) - \log P_f(\mathbf{y}_{2*}|\mathbf{x})\right)^2, \quad \Phi_f^K(\mathbf{x}) = \left(\log P_f(\mathbf{y}_{1*}^K|\mathbf{x}) - \log P_f(\mathbf{y}_{2*}^K|\mathbf{x})\right)^2,
\tag{54}
$$

where $\mathbf{y}_{1*}, \mathbf{y}_{2*}$ are the top-2 outputs over the entire output space $\mathcal{V}^{N_r}$, and $\mathbf{y}_{1*}^K, \mathbf{y}_{2*}^K$ are the top-2 outputs in $\mathcal{Y}_K$. Define:

$$
\Delta_\infty(\mathbf{x}) = \log P_f(\mathbf{y}_{1*}|\mathbf{x}) - \log P_f(\mathbf{y}_{2*}|\mathbf{x}), \quad \Delta_K(\mathbf{x}) = \log P_f(\mathbf{y}_{1*}^K|\mathbf{x}) - \log P_f(\mathbf{y}_{2*}^K|\mathbf{x}),
\tag{55}
$$

so that $\Phi_f^\infty(\mathbf{x}) = (\Delta_\infty(\mathbf{x}))^2$, $\Phi_f^K(\mathbf{x}) = (\Delta_K(\mathbf{x}))^2$, and the error is:

$$
|\Phi_f^K(\mathbf{x}) - \Phi_f^\infty(\mathbf{x})| = |(\Delta_K(\mathbf{x}))^2 - (\Delta_\infty(\mathbf{x}))^2| = |\Delta_K(\mathbf{x}) - \Delta_\infty(\mathbf{x})| \cdot |\Delta_K(\mathbf{x}) + \Delta_\infty(\mathbf{x})|.
\tag{56}
$$

By Assumption B.2, $|\Delta_\infty(\mathbf{x})| \leq R_K(\mathbf{x})$, where $R_K(\mathbf{x}) = \log P_f(\mathbf{y}_{1*}^K|\mathbf{x}) - \min_{\mathbf{y} \in \mathcal{Y}_K} \log P_f(\mathbf{y}|\mathbf{x})$ is the log-likelihood diameter of $\mathcal{Y}_K$. Also, $|\Delta_K(\mathbf{x})| \leq R_K(\mathbf{x})$, so:

$$
|\Delta_K(\mathbf{x}) + \Delta_\infty(\mathbf{x})| \leq |\Delta_K(\mathbf{x})| + |\Delta_\infty(\mathbf{x})| \leq 2R_K(\mathbf{x}).
\tag{57}
$$

Thus, the error is bounded by:

$$
|\Phi_f^K(\mathbf{x}) - \Phi_f^\infty(\mathbf{x})| \leq |\Delta_K(\mathbf{x}) - \Delta_\infty(\mathbf{x})| \cdot 2R_K(\mathbf{x}).
\tag{58}
$$

We compute the expectation:

$$
\mathbb{E}[|\Phi_f^K(\mathbf{x}) - \Phi_f^\infty(\mathbf{x})|] \leq 2R_K(\mathbf{x}) \cdot \mathbb{E}[|\Delta_K(\mathbf{x}) - \Delta_\infty(\mathbf{x})|].
\tag{59}
$$

Define $Z = |\Delta_K(\mathbf{x}) - \Delta_\infty(\mathbf{x})|$. From the proof of Theorem 4.6 (Equation 43), Hoeffding's inequality gives:

$$
\Pr(Z \geq t) \leq 4\exp\left(-\frac{2Kt^2}{R_K^2(\mathbf{x})}\right).
\tag{60}
$$

The expectation of $Z$ is:

$$
\mathbb{E}[Z] = \int_0^\infty \Pr(Z \geq t)\,dt \leq \int_0^\infty 4\exp\left(-\frac{2Kt^2}{R_K^2(\mathbf{x})}\right)dt.
\tag{61}
$$

Substitute $u = \frac{2Kt^2}{R_K^2(\mathbf{x})}$, so $t = R_K(\mathbf{x})\sqrt{\frac{u}{2K}}$, $dt = \frac{R_K(\mathbf{x})}{2\sqrt{2K}\sqrt{u}}\,du$. Then:

$$
\mathbb{E}[Z] \leq \int_0^\infty 4e^{-u} \cdot \frac{R_K(\mathbf{x})}{2\sqrt{2K}\sqrt{u}}\,du = \frac{2R_K(\mathbf{x})}{\sqrt{2K}}\int_0^\infty \frac{e^{-u}}{\sqrt{u}}\,du.
\tag{62}
$$

Since $\int_0^\infty \frac{e^{-u}}{\sqrt{u}}\,du = \Gamma\left(\frac{1}{2}\right) = \sqrt{\pi}$, we have:

$$
\mathbb{E}[Z] \leq \frac{2R_K(\mathbf{x})}{\sqrt{2K}} \cdot \sqrt{\pi} = R_K(\mathbf{x})\sqrt{\frac{2\pi}{K}}.
\tag{63}
$$

Thus:

$$\mathbb{E}[|\Phi_f^K(\mathbf{x}) - \Phi_f^\infty(\mathbf{x})|] \leq 2R_K(\mathbf{x}) \cdot R_K(\mathbf{x})\sqrt{\frac{2\pi}{K}} = 2R_K^2(\mathbf{x})\sqrt{\frac{2\pi}{K}}. \tag{64}$$

To account for event $B = \{\mathbf{y}_{1*} \notin \mathcal{Y}_K \text{ or } \mathbf{y}_{2*} \notin \mathcal{Y}_K\}$ with $\Pr(B) \leq 2\varepsilon_{\text{tail}}$ (from Lemma B.1 and union bound), we note that on event $A = \{\mathbf{y}_{1*} \in \mathcal{Y}_K \text{ and } \mathbf{y}_{2*} \in \mathcal{Y}_K\}$, the error is zero. Thus, we add a conservative term for event $B$, where the error is at most $2R_K^2(\mathbf{x})$ (since $|\Delta_K|, |\Delta_\infty| \leq R_K(\mathbf{x})$, so $|\Phi_f^K(\mathbf{x}) - \Phi_f^\infty(\mathbf{x})| \leq 2R_K^2(\mathbf{x})$):

$$\mathbb{E}[|\Phi_f^K(\mathbf{x}) - \Phi_f^\infty(\mathbf{x})| \cdot \mathbb{1}_B] \leq 2R_K^2(\mathbf{x}) \cdot \Pr(B) \leq 2R_K^2(\mathbf{x}) \cdot 2\varepsilon_{\text{tail}} = 4R_K^2(\mathbf{x})\varepsilon_{\text{tail}}, \tag{65}$$

where $\mathbb{1}_B$ is the indicator function which is 1 only when event $B$ occurs.

Combining both terms, the expected error is:

$$\mathbb{E}[|\Phi_f^K(\mathbf{x}) - \Phi_f^\infty(\mathbf{x})|] \leq 2R_K^2(\mathbf{x})\sqrt{\frac{2\pi}{K}} + 4R_K^2(\mathbf{x})\varepsilon_{\text{tail}}, \tag{66}$$

where $\varepsilon_{\text{tail}} = \left(1 - P_f(\mathbf{y}_{1*}^K|\mathbf{x})\right)^K$. This completes the proof. $\qquad\square$

### B.7  PROOF OF COROLLARY 4.8

*Proof.* We aim to bound the tail probability $\Pr(|\Phi_f^K(\mathbf{x}) - \Phi_f^\infty(\mathbf{x})| \geq \lambda)$ for $\lambda > 0$. Using the same notation as in Theorem 4.7, we have:

$$|\Phi_f^K(\mathbf{x}) - \Phi_f^\infty(\mathbf{x})| = |\Delta_K(\mathbf{x}) - \Delta_\infty(\mathbf{x})| \cdot |\Delta_K(\mathbf{x}) + \Delta_\infty(\mathbf{x})|. \tag{67}$$

Since $|\Delta_K(\mathbf{x}) + \Delta_\infty(\mathbf{x})| \leq 2R_K(\mathbf{x})$, let $Z = |\Delta_K(\mathbf{x}) - \Delta_\infty(\mathbf{x})|$, so:

$$|\Phi_f^K(\mathbf{x}) - \Phi_f^\infty(\mathbf{x})| \leq Z \cdot 2R_K(\mathbf{x}). \tag{68}$$

Thus:

$$\Pr(|\Phi_f^K(\mathbf{x}) - \Phi_f^\infty(\mathbf{x})| \geq \lambda) = \Pr(Z \cdot 2R_K(\mathbf{x}) \geq \lambda) = \Pr\left(Z \geq \frac{\lambda}{2R_K(\mathbf{x})}\right). \tag{69}$$

From the proof of Theorem 4.6 (Equation 43), Hoeffding's inequality gives:

$$\Pr(Z \geq t) \leq 4\exp\left(-\frac{2Kt^2}{R_K^2(\mathbf{x})}\right). \tag{70}$$

Set $t = \frac{\lambda}{2R_K(\mathbf{x})}$:

$$\Pr\left(Z \geq \frac{\lambda}{2R_K(\mathbf{x})}\right) \leq 4\exp\left(-\frac{2K \cdot \left(\frac{\lambda}{2R_K(\mathbf{x})}\right)^2}{R_K^2(\mathbf{x})}\right) = 4\exp\left(-\frac{K\lambda^2}{2R_K^4(\mathbf{x})}\right). \tag{71}$$

Define events $A = \{\mathbf{y}_{1*} \in \mathcal{Y}_K \text{ and } \mathbf{y}_{2*} \in \mathcal{Y}_K\}$ and $B = \{\mathbf{y}_{1*} \notin \mathcal{Y}_K \text{ or } \mathbf{y}_{2*} \notin \mathcal{Y}_K\}$. On event $A$, the error is zero, so it does not contribute to the tail probability. On event $B$, with $\Pr(B) \leq 2\varepsilon_{\text{tail}}$ (from Lemma B.1 and union bound), the tail probability is bounded by:

$$\Pr(|\Phi_f^K(\mathbf{x}) - \Phi_f^\infty(\mathbf{x})| \geq \lambda) \leq \Pr\left(\left\{Z \geq \frac{\lambda}{2R_K(\mathbf{x})}\right\} \cap B\right) + \Pr(A). \tag{72}$$

Since $\Pr(A) \geq 1 - 2\varepsilon_{\text{tail}}$ and the error is zero on $A$, we focus on event $B$:

$$\Pr\left(\left\{Z \geq \frac{\lambda}{2R_K(\mathbf{x})}\right\} \cap B\right) \leq \Pr\left(Z \geq \frac{\lambda}{2R_K(\mathbf{x})}\right) + \Pr(B) \leq 4\exp\left(-\frac{K\lambda^2}{2R_K^4(\mathbf{x})}\right) + 2\varepsilon_{\text{tail}}. \tag{73}$$

Thus, the tail probability is:

$$\Pr\left(|\Phi_f^K(\mathbf{x}) - \Phi_f^\infty(\mathbf{x})| \geq \lambda\right) \leq 4\exp\left(-\frac{K\lambda^2}{2R_K^4(\mathbf{x})}\right) + 2\varepsilon_{\text{tail}}, \tag{74}$$

where $\varepsilon_{\text{tail}} = \left(1 - P_f(\mathbf{y}_{1*}^K|\mathbf{x})\right)^K$. This completes the proof. $\qquad\square$

## C  SUPPLEMENTAL EXPERIMENTS

### C.1  IMPLIPLICATIONS OF $K$-DPS

In this subsection, we provide several preliminary, proof-of-concept applications of the $K$-DPS algorithm.

#### C.1.1  ALIGNMENT

**Settings.** We select three groups of LLMs in our experiments with each group containing two models before and after the human alignment. We respectively visualize the linear interpolation visualization, nearest interpolation visualization, heatmap, and three dimensional visualization for each model, as shown in each row. The color indicates the value of corresponding positions, where a darker color indicates more smaller $K$-DPS score. We use $K = 2,500$ in our experiments, and the input queries come from AdvBenchk (Zou et al., 2023).

**Results.** As shown in Figure 6, the decision boundary of aligned models becomes dramatically smoother and flatter compared to their pre-alignment counterparts when evaluated on adversarial prompts from AdvBench. This striking smoothing effect indicates that alignment substantially reduces regions of high confidence in harmful outputs (i.e., sharp peaks with high K-DPS scores). Instead, it creates broad, low-K-DPS basins that strongly favor refusal. This geometric transformation directly explains both: *i) Why jailbreaks succeed on unaligned models*: they target narrow, high-confidence "vulnerability spikes" that remain in the pre-alignment landscape; *ii) Why alignment mitigates most jailbreaks*: it eliminates these spikes entirely, making harmful responses probabilistically unlikely across vast regions of prompt space.

Moreover, by applying the same K-DPS visualization to intermediate checkpoints throughout the alignment process, we can even track precisely how these dangerous peaks progressively flatten and how decision boundaries move, offering the first fine-grained geometric view of how safety training reshapes the model's decision manifold step by step.

This level of mechanistic explanation and dynamic visualization was previously infeasible with prior probing or interpretability techniques, but becomes straightforward and highly revealing through our DPS-based decision boundary construction.

#### C.1.2  UNLEARNING

**Settings.** K-DPS also enables fine-grained analysis of machine unlearning algorithms, where prior interpretability tools offer almost no intuitive explanations.

We apply our K-DPS to two representative unlearning methods: Gradient Ascent (GA) (Yao et al., 2024) and Negative Preference Optimization (NPO) (Zhang et al., 2024). We use the standard Harry Potter book as the forget (unlearning) corpus and a Wikipedia subset[2] as the retain set. During unlearning, we continue training on the retain set using standard gradient descent (GDR) or the KL divergence (KLR) from the original model. We set K = 2,000.

**Results.** As shown in Figure 7, our K-DPS visualizations provide a far clearer picture of the side effects of unlearning than previously possible (Liu et al., 2025a; Geng et al., 2025). Prior work could only report that unlearning without proper retention training degrades overall performance, yet was unable to show what form this degradation takes in the model's internal decision process. With K-DPS, we reveal that naïve unlearning methods (e.g., pure Gradient Ascent) can trigger catastrophic collapse of the entire decision manifold: large portions of the prompt space that were previously smooth become extremely jagged and fragmented, with erratic high- and low-K-DPS spikes appearing in regions unrelated to the forget corpus.

In contrast, when retention training is included (e.g., GA-GDR and GA-KLR), the damage is substantially mitigated. Among these, GA-KLR (KL-regularized retention) preserves a decision surface that is visibly the closest to the original model, albeit still perceptibly distorted. This aligns with quantitative results in the unlearning literature showing that KL regularization best preserves general capabilities.

---

[2]https://huggingface.co/datasets/rag-datasets/rag-mini-wikipedia

The above observations demonstrate that K-DPS not only confirms known phenomena at a qualitative level but, for the first time, makes the geometric nature of "unlearning damage" directly observable and comparable across methods.

**Others.** Beyond the two core insights above, we believe our decision-boundary framework can naturally extend to a wide range of important LLM phenomena that have so far resisted precise mechanistic analysis, including: *i)* the precise locations and shapes of jailbreak vulnerabilities in prompt space, *ii)* the emergence and structure of memorization regions, *iii)* the geometry of hallucination-prone areas versus high-fidelity regions, *iv)* systematic changes in the boundary during continual learning or catastrophic forgetting, and so on.

## C.2    DISCUSSIONS

### C.2.1    WILL THE TWO HIGHEST-PROBABILITY SEQUENCES OFTEN BE NEARLY IDENTICAL?

In this subsection, we dive into the analysis of the sampled responses under $K$-DPS, where a potential issue is that the top-2 completions might be extremely similar under some situations, sometimes even differing by only a single token or a minor lexical variation, which may question the effectiveness of the proposed method.

In this subsection, we examine the sampled completions under $K$-DPS and address a potential concern: in certain cases, the top-2 generations can be nearly identical, differing by only a single token, minor lexical variations, or superficial formatting. Such high surface-level similarity might raise questions about whether K-DPS is sufficiently sensitive to meaningful semantic discrepancies.

Suggested by the visualizations (Figures 10, 6, 7, 11, and 4), the fraction of such near-identical cases is actually not significant in practice, especially in the regions that matter most for decision-boundary analysis. As clearly demonstrated in these figures, meaningful and sharp decision boundaries consistently appear. Degenerate boundaries (which would occur if top-1 and top-2 were almost always identical) are rarely observed across the vast majority of inputs we study, indicating that this is not a common phenomena in the sampling.

To further investigate this problem, we conduct some quantitative analysis, which confirms substantial lexical and semantic divergence in K-DPS construction. Specifically, for all candidate sequences used to construct Figure 10, we measured the normalized Levenshtein edit distance between top-1 and top-2 completions.

| K-DPS Score Range | Avg. Normalized Edit Distance |
|---|---|
| $< 0.1$ | 0.15 |
| $< 0.5$ | 0.20 |
| $< 1.0$ | 0.23 |
| $< 5.0$ | 0.30 |
| $< 10.0$ | 0.32 |

Table 1: Average normalized edit distance between top-1 and top-2 completions as a function of $K$-DPS confidence score range.

As shown in Table 1, even in the highest-confidence regime ($K$-DPS values ¡ 0.1), the top-2 sequences differ by  15% of tokens on average; near decision boundaries (higher $K$-DPS), divergence reaches 30–32%, which is far from trivial variants and typically meaningful.

Regarding the situation that the edit distance between top-2 sequences are small, we can address it with some simple filtering strategies, such as slightly increasing the sampling temperature during candidate generation or directly discarding pairs with near-zero edit distance.

### C.2.2    $K$-DPS VERSUS MODEL UNCERTAINTY

We notice that the construction of explicit decision boundaries in the representation space might exhibit connections with several core research areas in LLMs, particularly confidence estimation and uncertainty quantification (UQ) (Geng et al., 2024; Huang et al., 2025; Liu et al., 2025b; Xia et al., 2025b; Shorinwa et al., 2025; Lin et al., 2024). These uncertainty quantification approaches

typically include verbalized confidence expressed in natural language (Kadavath et al., 2022), token-level entropy of the output distribution (Kuhn et al., 2023), and semantic entropy computed over semantically equivalent clusters of multiple generations (Kuhn et al., 2023; Farquhar et al., 2024), with the latter achieving state-of-the-art performance in hallucination detection and selective generation tasks.

While these methods also measure the certainty and the confidence of the model decision, our K-DPS decision-boundary construction differs from them in several fundamental aspects:

First, classical uncertainty quantification techniques Kadavath et al. (2022); Kuhn et al. (2023); Farquhar et al. (2024) are essentially heuristic or sampling-based scores lacking formal theoretical guarantees, whereas K-DPS provides provably conservative classification boundaries with explicit error bounds. It achieves a precise and meaningful approximation of the decision boundary. Second, existing UQ methods operate at the instance level and treat each generation independently, while K-DPS explicitly builds and reasons over distribution-level decision boundaries, enabling global geometric understanding of the model's reliable support. For the usage, conventional approaches remain largely oblivious to the location of samples relative to the empirical data manifold, whereas K-DPS deliberately identifies and penalizes anomalous boundary samples that fall near or outside the observed support of each semantic class. These distinctions shift the paradigm from post-hoc uncertainty scoring to principled, boundary-aware certification of LLM generations.

Nevertheless, we acknowledge that K-DPS and traditional uncertainty quantification methods indeed share some core insights. Both paradigms ultimately aim to identify when an LLM's output is unreliable, whether due to hallucination, out-of-distribution inputs, adversarial attacks, or memorization-based spurious responses. Technically, they all ground their analysis in the same internal representations of the model: prior UQ approaches directly use raw logits, token probabilities, or hidden states to compute verbalized confidence or entropy measures, whereas K-DPS leverages the DPF as the theoretical indicator to perform boundary construction. Consequently, the decision boundary learned by $K$-DPS can be interpreted as a geometrically principled extension of uncertainty signals: samples assigned high semantic entropy or low verbalized confidence often naturally fall into low-density or boundary regions detected by $K$-DPS, providing a unified explanatory framework for why existing UQ methods succeed or fail on specific examples. In practice, the two families of approaches are highly complementary: uncertainty scores can serve as lightweight pre-filters, while K-DPS offers stricter, certifiable analysis for LLM inference.

### C.3 Empirical Error Analysis under Log Scales

We present log-log reconstructions of Figure 1 and Figure 2 in Figure 5 and Figure 8, respectively.

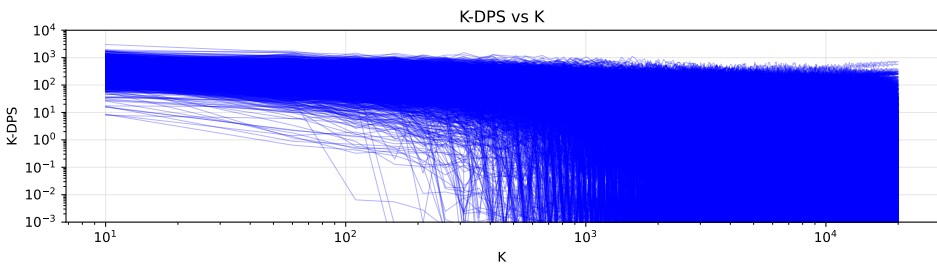

Figure 5: Effect of sampling size $K$ on the values of decision potential function, with each blue point representing the $K$-DPS value for a single input sample. Each blue line represents a trend of $K$-DPS for one input sample.

### C.4 Supplemental Visualization

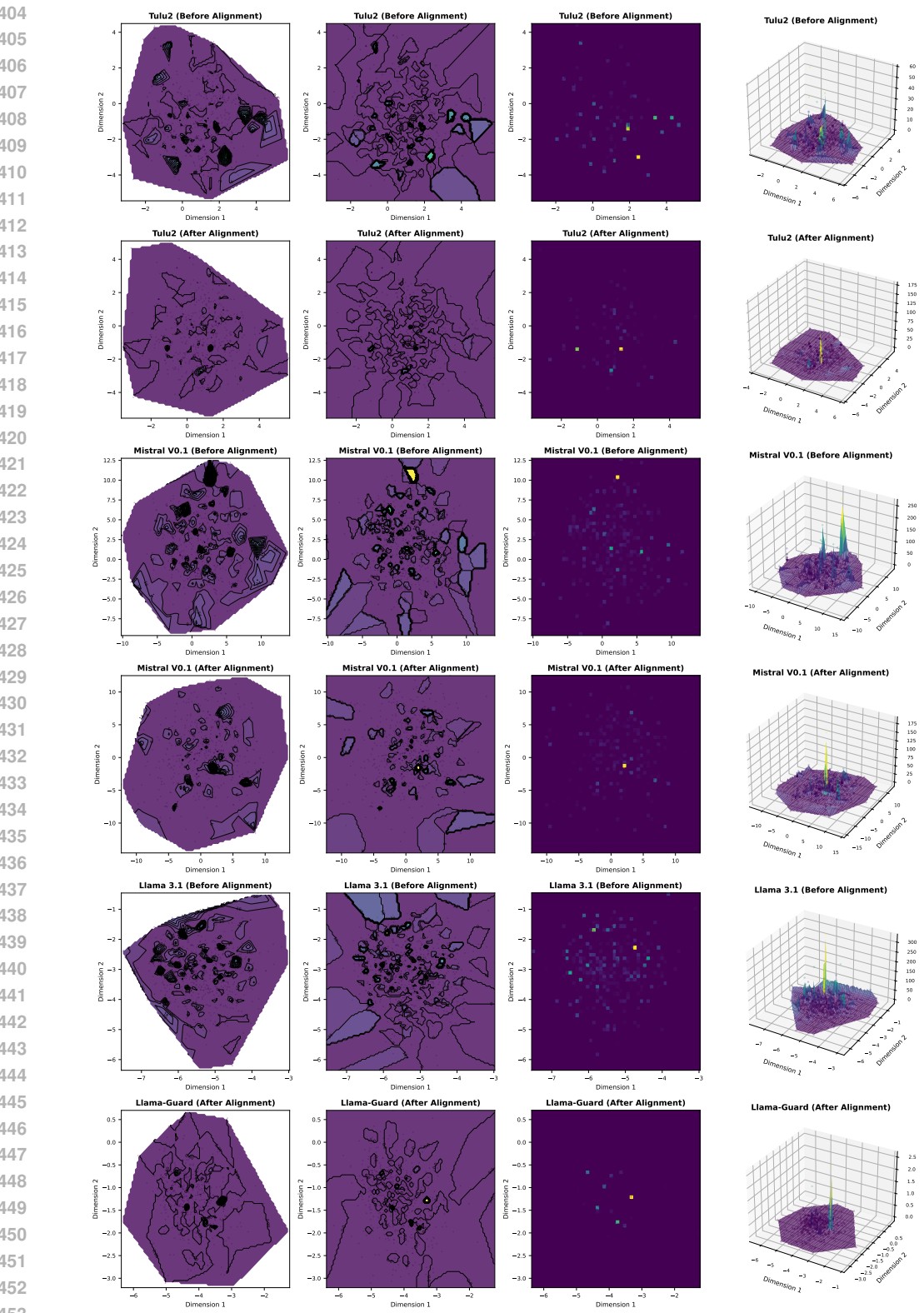

Figure 6: Visualization of decision boundaries constructed by K-DPS in the semantic embedding space, before and after preference alignment. Each row (top to bottom) presents results for Tulu-2-7B (SFT only), Tulu-2-7B-DPO, Mistral-7B-v0.1, Zephyr-7B, Meta-Llama-3.1-8B, and Llama-Guard-3-8B. Columns (left to right) display: 2D interpolation using linear and nearest-neighbor methods, heatmap, and 3D rendering of the decision boundary.

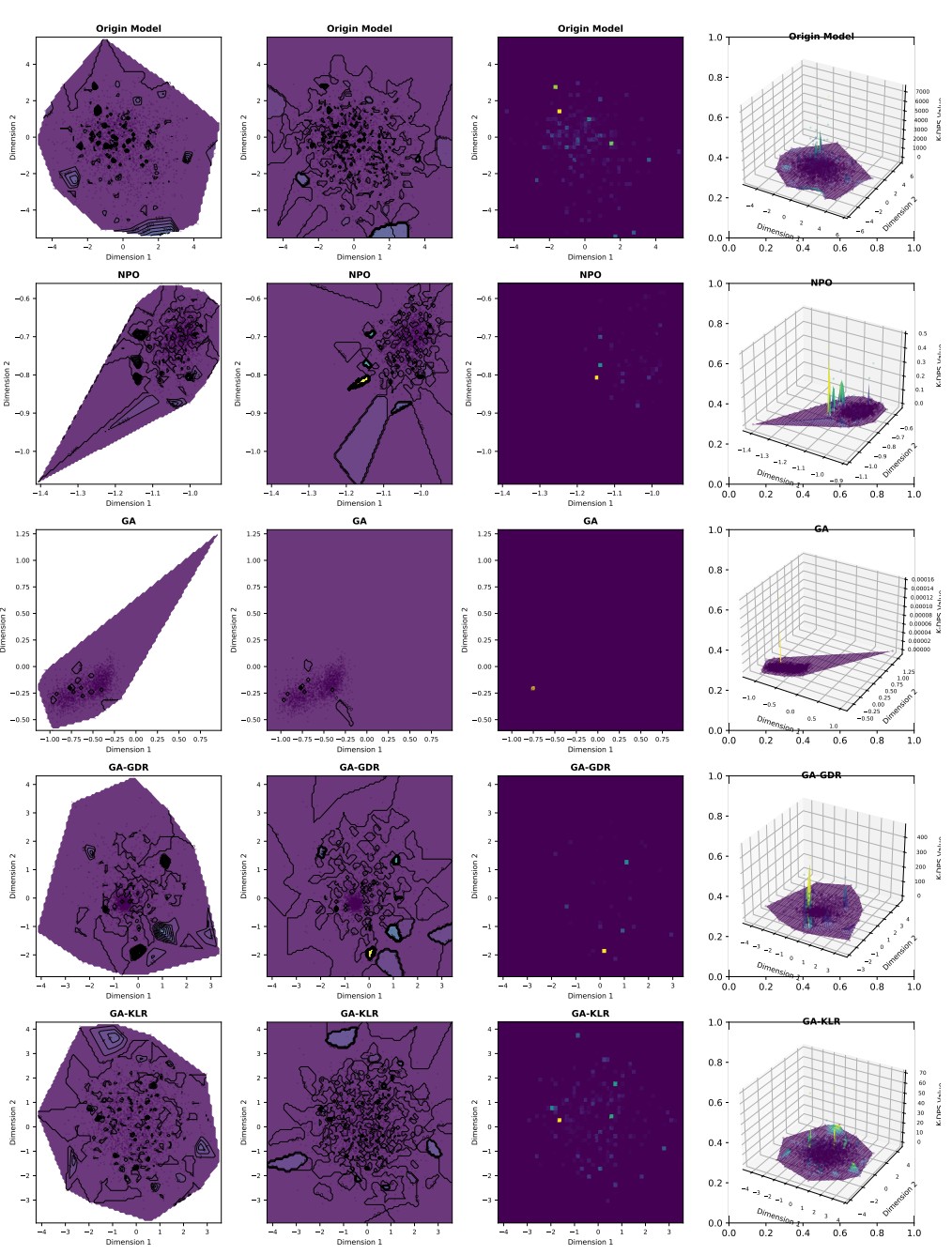

Figure 7: Visualization of decision boundaries constructed by K-DPS on machine unlearning models. Experimental settings and layout follow Figure 6; rows from top to bottom correspond to the original model and its unlearned variants using different unlearning algorithms.

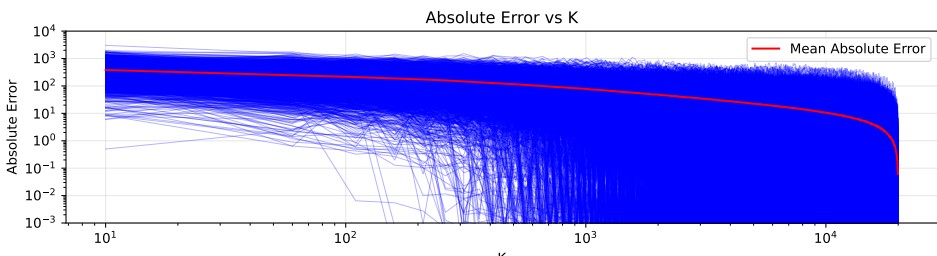

Figure 8: Effect of sampling size $K$ on the absolute error between the reference $K$-DPS (computed with $K = 20,000$) and $K$-DPS values for varying $K$. Each blue line represents a trend of absolute error across input samples.

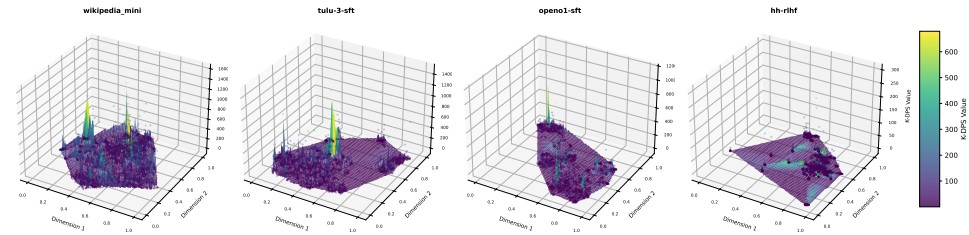

Figure 9: Three-dimensional visualization of the $K$-DPS ($K = 2,500$) for Llama-3.2-1B on four datasets.

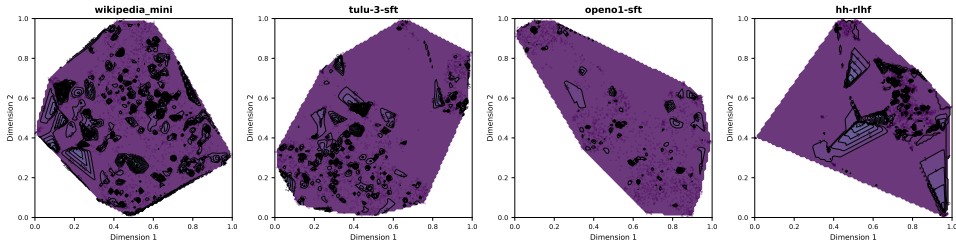

Figure 10: Contour visualization of 2,500-grained decision potential surface for Llama3.2-1B on four datasets with linear interpolation.



Figure 11: Contour visualization of 2,500-grained decision potential surface of Llama3.2-1B on four datasets with nearest interpolation.



Figure 12: Heatmap visualization of the decision potential surface on Four datasets.

