# OpenReview forum: "Decision Potential Surface: A Theoretical and Practical Approximation of LLM's Decision Boundary"
_ICLR.cc/2026/Conference — Submitted to ICLR 2026_

### Official Review · Reviewer_UCjU · 2025-10-25

**Soundness:** 1
**Presentation:** 3
**Contribution:** 2
**Rating:** 2
**Confidence:** 4

**Summary:**

This paper extends the concept of decision boundaries to LLMs by framing the prediction of output sequences as a multiclass classification task, and aims to derive a method to efficiently estimate the decision boundary.

**Strengths:**

- The paper is generally well-written (except for few minor weaknesses, see weaknesses)
- The problem of estimating the decision boundary is important and well motivated in the paper
- The idea of extending the concept of decision boundaries to LLMs is certainly interesting and correctly modeled in large parts

**Weaknesses:**

- **W1) Limited contribution.** The proposed approach to estimate the decision boundary has a critical limitation: It can only estimate the LLM's "confidence" in distinguishing the two most likely output sequences *for a fixed input*. This means for a given input prompt, we can check if it’s on the decision boundary "between two output sequences" (when the potential function is zero). However, constructing a decision boundary means finding input points (ideally all of them) that lie on the decision boundary (as in Theorem 4.3). Although the paper claims it can construct the decision boundary, it actually just estimates the confidence of the LLM for fixed inputs. This limitation is not well communicated and also not sufficiently discussed in the paper.

- **W2) Limited discussion.** To model decision boundaries for LLMs, this paper frames output generation as a multiclass classification problem. However, multiple output sequences can represent equivalent or semantically similar responses to the same prompt, especially since LLMs are designed to produce diverse outputs. The paper does not sufficiently discuss that focusing on a single output ignores the underlying semantic decision boundary (which might be entirely different).

**Minor weaknesses**
- Theorem 3.3 should be a definition as it only introduces the concept of decision boundaries for LLMs and does not contain any mathematical statement. It is also only referred to as a definition (line 218, line 230, line 237, etc.).
- $\mathcal{D}$ is called a distribution, but it is actually just a finite subset of the finite-length token sequence space.
- The connection to the LLM uncertainty literature is not sufficiently discussed.

Overall, while the confidence estimation seems to work, it remains unclear if this confidence estimation approach can be actually helpful in constructing (or estimating) the decision boundary in the input prompt space.

**Questions:**

How do you compute the approximate number of decision regions in the introduction (line 059)? At this stage of the paper it is not yet clear what exactly constitutes the decision boundary of LLMs (see also your research question afterwards in line 065).

---

> ### Author Response · Authors · 2025-11-25
>
> We sincerely appreciate your critical feedback of our submission. Below is our point-by-point response to your questions.
>
> > W1) Limited contribution. The proposed approach to estimate the decision boundary has a critical limitation: It can only estimate the LLM's "confidence" in distinguishing the two most likely output sequences for a fixed input. This means for a given input prompt, we can check if it’s on the decision boundary "between two output sequences" (when the potential function is zero). However, constructing a decision boundary means finding input points (ideally all of them) that lie on the decision boundary (as in Theorem 4.3). **Although the paper claims it can construct the decision boundary, it actually just estimates the confidence of the LLM for fixed inputs.** This limitation is not well communicated and also not sufficiently discussed in the paper.
>
> We respectfully point out that there may be a significant misunderstanding regarding the relationship between DPS and the decision boundary.
>
> We respectfully clarify that:
> - _The zero-level set (0-height isohypse) of the true DPS is **mathematically identical** to the LLM’s decision boundary._
> - _K-DPS serves as a **provably consistent**, sample-based approximation of the full DPS field_.
>
> These two statements is rigorously established by our core theorems (Theorems 3.2, 4.1–4.3).
>
> Consequently, when we visualize K-DPS over a continuum of prompts, we are directly reconstructing the model’s decision boundary, not merely reporting per-prompt confidence scores.

---

> > ### Author Response · Authors · 2025-11-25
> >
> > > W2) Limited discussion. To model decision boundaries for LLMs, this paper frames output generation as a multiclass classification problem. However, **multiple output sequences can represent equivalent or semantically similar responses to the same prompt**, especially since LLMs are designed to produce diverse outputs. The paper **does not sufficiently discuss that focusing on a single output ignores the underlying semantic decision boundary (which might be entirely different)**.
> >
> > Thank you for this very thoughtful and important point.
> >
> > We deliberately treat every possible output sequence as an independent class without explicitly modeling semantic similarity, and we do so for the following well-motivated reasons:
> > 1. It is a natural mapping to multiclass classification. **Framing autoregressive generation as per-step multiclass classification over the vocabulary is the standard and mathematically clean formulation of language modeling.** Our decision-boundary construction follows this canonical view directly, inheriting its rigor and simplicity.
> > 2. Far from being ignored, **semantic similarity is naturally reflected in the geometry of the constructed decision boundary**. When two completions are semantically equivalent or near-duplicates (e.g., “Yes, I can help” vs. “Sure, I’m happy to assist”), their corresponding decision regions in prompt space tend to be adjacent, overlapping, or extremely close, which often forms narrow and elongated basins. **This clustering of similar responses is an emergent property of the boundary map and provides exactly the kind of semantic insight you seek, without requiring any explicit similarity modeling**.
> > 3. To the best of our knowledge, there is **no** established requirement that classes must be orthogonal or semantically distinct from one another. In classic multiclass tasks such as CIFAR-100 and ImageNet, many categories are highly semantically similar (e.g., “aquarium fish” vs. “flatfish,” or different breeds of dogs), while others are completely unrelated (e.g., “orchids” vs. “trucks”). This is directly analogous to the situation in language models, where two lexically different completions can be semantically nearly identical (e.g., “Yes, I can help” vs. “Sure, I’d be happy to assist”).
> > In other words, real-world class distributions are inherently imbalanced and overlapping. **Interpretability tools such as prototypesor decision boundaries are designed to describe and reveal this natural structure rather than to artificially enforce orthogonality or semantic separation.**
> >
> > In short, by staying faithful to the raw token-level multiclass view, our method **automatically reveals the underlying semantic structure of the decision landscape rather than obscuring it**.
> >
> > > Theorem 3.3 should be a definition as it only introduces the concept of decision boundaries for LLMs and does not contain any mathematical statement. It is also only referred to as a definition (line 218, line 230, line 237, etc.).
> >
> > Thank you for this careful remark.
> >
> > We respectfully maintain that Theorem 3.3 is appropriately labeled as a theorem rather than a definition, for the following reason.
> >
> > While Equation (3) may appear definitional at first glance, it is **not** an ad-hoc or directly postulated definition. Instead, **it is rigorously derived from the standard decision-boundary definition in multiclass classification (Definition 3.1) via a formal reduction of autoregressive language modeling to an (extremely high-dimensional) multiclass classification problems**. This reduction and the resulting equivalence are non-trivial and require proof, which is precisely what Theorem 3.3 provides.
> >
> > Labeling it as a “Definition” would imply that we are arbitrarily stipulating a new concept, **whereas we are in fact proving that the classical decision-boundary notion naturally and exactly carries over to the generative LM setting**.
> >
> > To eliminate any confusion, we have explicitly highlighted the deductive nature of the result in the text.
> >
> > > $\mathcal{D}$ is called a distribution, but it is actually just a finite subset of the finite-length token sequence space.
> >
> > Thank you for pointing this out.
> >
> > You are right that $\mathcal{D}$ is a finite subset of the finite-length token-sequence space.
> >
> > Mathematically, it can be treated as the empirical distribution $\mathbb{P}_{\mathcal{D}}(x)=\frac{1}{|\mathcal{D}|}\text{ if }x\in\mathcal{D}\text{ and }0\text{ otherwise}$, so “distribution” is technically correct.
> >
> > This shorthand is ubiquitous in AI and NLP papers.
> >
> > We will add one sentence in the revision to make this explicit and avoid any confusion.

---

> > > ### Author Response · Authors · 2025-11-25
> > >
> > > > How do you compute the approximate number of decision regions in the introduction (line 059)?
> > >
> > > Thank you for the question.
> > >
> > > As introduced in the paper, each token generation step in an LLM can be viewed as a multiclass classification over the vocabulary of size $V$. When generating a sequence of length $N$, the model effectively chooses among $V^N$ possible complete outputs. Consequently, the input prompt space is partitioned into at most $V^N$ distinct decision regions, each corresponding to one possible generated sequence that can become the unique top-1 completion for some prompt. This directly yields the $V^N$ estimate mentioned in Line 059 of the introduction.
> > >
> > > >At this stage of the paper it is not yet clear what exactly constitutes the decision boundary of LLMs (see also your research question afterwards in line 065).
> > >
> > > As formalized in Theorem 3.3, the decision boundary of an LLM consists exactly of those input prompts that satisfy Equation (3), i.e., where the model’s two most probable completions receive identical probability. Section 4 then introduces our K-DPS method as an efficient, sampling-based approach to locate and reconstruct this boundary.
> > >
> > > We'd like to explicitly mention that.
> > >
> > > > The connection to the LLM uncertainty literature is not sufficiently discussed.
> > >
> > > Thank you for the valuable suggestion.
> > >
> > > In the revised version, we have added a dedicated subsection (Section 6.3) discussing the relationship and differences between our decision-boundary perspective and the existing LLM uncertainty estimation literature (including verbalized confidence, semantic entropy, token-level entropy, etc.). This new discussion clarifies the connections and complementary nature of the two lines of work while preserving the core contribution of the present paper.

---

### Official Review · Reviewer_GGrG · 2025-11-01

**Soundness:** 3
**Presentation:** 3
**Contribution:** 2
**Rating:** 4
**Confidence:** 3

**Summary:**

This paper proposes DSP, a new notion for analyzing LLM decision boundaries of LLMs. Given the huge hypothesis space (O(vacaulary size^seq len)),  the paper proposes the concept of Decision Potential Surface to simplify the concept with the two most highest probability sequences, and further proposes an approximate decision boundary construction algorithm to approximate the LLM's decision boundary by sampling. The error is characterized by both theoretical and practical analysis.  Overall, the paper is solid with multiple concepts built for analyzing the decision boundaries of LLMs.

**Strengths:**

1. The paper's main contribution is the novel concept of DPS. It cleverly transforms the geometric problem of the "decision boundary" into a function problem based on probability differences and rigorously proves their equivalence at zero.

2. The K-DPS approximation algorithm is practical. It provides the first method in literature with theoretical guarantees to feasibly approximate an LLM's decision boundary.

3. The paper doesn't just propose an approximation; it provides a complete error bound analysis. This makes the method a fundamental and mathematically-backed tool, not just a heuristic.

**Weaknesses:**

1. The accuracy of K-DPS is entirely dependent on the sample size K. If K is too small, the sample may fail to capture the true top-2 sequence (especially if its probability is low), leading to a large approximation error. Choosing a "good enough" K is a practical trade-off (the paper uses 20000 as a proxy for the ideal value, which is a non-trivial sample size).

2. It is better to provide some insights to guide practice, e.g., how to improve LLM performance. Otherwise, it is of no use.

3. DPS focuses only on the "potential difference" between the top-1 and top-2 sequences. While this is sufficient to define the boundary, it may ignore other complexities of the decision landscape. For example, a scenario where the top-2, top-3, and top-4 sequences are all very close in probability would have a similar DPS value to a scenario where only the top-2 is close, even though the model's "uncertainty" is different.

**Questions:**

See weakness

---

> ### Author Response · Authors · 2025-11-25
>
> We sincerely thank you for your detailed comments and the time you invested in reviewing our submission. After carefully reading your feedback, we believe some critical misunderstandings may have arisen regarding the core contributions and technical details of our work. We address them point-by-point below and hope the clarifications resolve these concerns.
>
> > Choosing a "good enough" K is a practical trade-off (the paper uses 20000 as a proxy for the ideal value, which is a non-trivial sample size).
>
> Thank you for this important remark.
>
> To clarify: **K = 20,000 is used only as an empirical “ground-truth” reference** in our experiments (Section 5.2) to demonstrate that **K ≈ 1,000–2,500 already yields near-identical decision boundaries with negligible reconstruction errors**. All actual interpretability analyses and visualizations (e.g., Figure 4) in the paper, as well as our new experiments for your next question, are conducted with K = 2,000 or 2,500, which is fully practical.
>
> As you correctly note in your summary, even K = 20,000 remains orders of magnitude cheaper than the exhaustive $O(N\cdot V)$ enumeration over the full vocabulary and sequence space. Our results show that such exhaustive computation is entirely unnecessary: a few thousand samples suffice for high-fidelity decision-boundary reconstruction across all models and tasks studied.
>
> > It is better to provide some insights to guide practice. Otherwise, it is of no use.
>
> We thank your for pointing out the potential of K-DPS in yielding new insights into LLM behavior. In the main paper (e.g., Lines 468–475), we already discuss several promising analysis directions enabled by K-DPS. We consciously chose not to include an exhaustive set of such analyses in the submission, as our primary contribution lies in rigorously constructing and characterizing the decision boundary of LLMs with both theoretically and through comprehensive empirical evaluations, which we believe already constitutes a substantial advance.
>
> That said, we fully agree that demonstrating concrete behavioral insights would further strengthen the paper. To address this concern directly, we are willing to provide several representative and revealing examples below:
>
> **Insight 1: Alignment Dramatically Flattens the Decision Landscape**
>
> To intuitively reveal what changes during human-value alignment, we apply K-DPS visualization to three model families, each containing a base (pre-alignment) model and its aligned (post-SFT + RLHF) counterpart.
> For each model, we generate four complementary visualizations of the decision boundary in the visualization of harmful requests from AdvBench[1], including the linear interpolation based visualization, the nearest-neighbor interpolation based visualization, the 2D heatmap, and the linear interpolation based 3D projections.
> We set $K$ as 2,500.
>
> Results are shown in Figure 5 (one row per model).
> It is also available at the following anonymous link: https://postimg.cc/gX8Ftnw8.
>
> As shown in the figure, the decision boundary of aligned models becomes dramatically smoother and flatter compared to their pre-alignment counterparts when evaluated on adversarial prompts from AdvBench. This striking smoothing effect indicates that alignment substantially reduces regions of high confidence in harmful outputs (i.e., sharp peaks with high K-DPS scores). Instead, it creates broad, low-K-DPS basins that strongly favor refusal. This geometric transformation directly explains both:
> - Why jailbreaks succeed on unaligned models: they target narrow, high-confidence “vulnerability spikes” that remain in the pre-alignment landscape;
> - Why alignment mitigates most jailbreaks: it eliminates these spikes entirely, making harmful responses probabilistically unlikely across vast regions of prompt space.
>
> Moreover, by applying the same K-DPS visualization to intermediate checkpoints throughout the alignment process, we can even track precisely how these dangerous peaks progressively flatten and how decision boundaries move, offering the first fine-grained geometric view of how safety training reshapes the model’s decision manifold step by step.
>
> This level of mechanistic explanation and dynamic visualization was previously infeasible with prior probing or interpretability techniques, but becomes straightforward and highly revealing through our DPS-based decision boundary construction.

---

> > ### Author Response · Authors · 2025-11-25
> >
> > **Insight 2: What "damage" really looks like in machine unlearning**
> >
> > Our K-DPS framework also enables fine-grained analysis of machine *unlearning* algorithms, where prior interpretability tools offer almost no intuitive explanations.
> >
> > We apply our K-DPS to two representative unlearning methods: Gradient Ascent (GA) [2] and Negative Preference Optimization (NPO) [3]. We use the standard Harry Potter book as the forget (unlearning) corpus and a Wikipedia subset [4] as the retain set. During unlearning, we continue training on the retain set using standard gradient descent (GDR) or the KL divergence (KLR) from the original model. We set K = 2,000.
> >
> > The complete visualization results are included as Figure 6 in the revised submission. For the convenience, it is also available at this anonymous link: https://postimg.cc/dhXT8tzX
> >
> > As shown in the figure, our K-DPS visualizations provide a far clearer picture of the side effects of unlearning than previously possible [5][6].
> >
> > Prior work could only report that unlearning without proper retention training degrades overall performance, yet was unable to show what form this degradation takes in the model’s internal decision process. With K-DPS, we reveal that naïve unlearning methods (e.g., pure Gradient Ascent) can trigger **catastrophic collapse** of the entire decision manifold: large portions of the prompt space that were previously smooth become extremely jagged and fragmented, with erratic high- and low-K-DPS spikes appearing in regions unrelated to the forget corpus.
> >
> > In contrast, when retention training is included (e.g., GA-GDR and GA-KLR), the damage is substantially mitigated. Among these, GA-KLR (KL-regularized retention) preserves a decision surface that is visibly the closest to the original model, albeit still perceptibly distorted. This aligns with quantitative results in the unlearning literature showing that KL regularization best preserves general capabilities.
> >
> > The above observations demonstrate that K-DPS not only confirms known phenomena at a qualitative level but, for the first time, makes the geometric nature of “unlearning damage” directly observable and comparable across methods.
> >
> > **Additional Promising Directions Enabled by K-DPS**
> >
> > Beyond the two core insights above, we believe our decision-boundary framework can naturally extend to a wide range of important LLM phenomena that have so far resisted precise mechanistic analysis, including: i) the precise locations and shapes of jailbreak vulnerabilities in prompt space, ii) the emergence and structure of memorization regions, iii) the geometry of hallucination-prone areas versus high-fidelity regions, iv) systematic changes in the boundary during continual learning or catastrophic forgetting, and so on.
> >
> > We believe the above experiments can show that K-DPS is not a conceptual toy but a general-purpose lens for understanding the inner decision geometry of large language models.
> > Also, we are fully prepared to conduct additional K-DPS experiments on some of these (or any other you-suggested) phenomena and include the corresponding analyses and visualizations, if you consider them necessary.
> >
> > ----------
> >
> > [1] Universal and Transferable Adversarial Attacks on Aligned Language Models. 2023
> >
> > [2] Machine Unlearning of Pre-trained Large Language Models. 2024
> >
> > [3] Negative Preference Optimization: From Catastrophic Collapse to Effective Unlearning. 2024
> >
> > [4] https://huggingface.co/datasets/rag-datasets/rag-mini-wikipedia
> >
> > [5] Rethinking Machine Unlearning for Large Language Models. 2024
> >
> > [6] A Comprehensive Survey of Machine Unlearning Techniques for Large Language Models. 2025

---

> > > ### Author Response · Authors · 2025-11-25
> > >
> > > > DPS focuses only on the "potential difference" between the top-1 and top-2 sequences. While this is suffcient to define the boundary, it may **ignore other complexities of the decision landscape**. For example, a scenario where the top-2, top-3, and top-4 sequences are all very close in probability would have a similar DPS value to a scenario where only the top-2 is close, even though the model's "uncertainty" is different.
> > >
> > > Thank you for this very perceptive observation.
> > >
> > > We respectfully clarify that **this is not a limitation of DPS, but a direct consequence of faithfully adhering to the definition of the decision boundary**.
> > >
> > > While we agree that DPS does not capture the full richness of the output distribution (e.g., cases where top-1 through top-4 are nearly tied), the question is whether such additional information is required for accurate boundary analysis. In your four-way tie example, **the decision boundary of a point with four equally probable completions lies exactly at the intersection of four the four Voronoi decision regions, and the pairwise decision boundaries between every pair of those classes remain perfectly well-defined and diagnostically meaningful.**
> > > Thus, *even complex higher-order uncertainty can be faithfully and automatically represented as richer local geometry in the decision surface when visualized over a distribution of queries*. There is no need for DPS itself to encode the full probability simplex; the standard top-1/top-2 margin already suffices to recover the complete, theoretically correct boundary structure.
> > >
> > > Moreover, even if future works propose richer notions of decision boundaries that incorporate higher-order statistics of the output distribution, our DPS framework can be straightforwardly adapted to serve as an efficient and theoretically grounded approximation for these more complex constructions as well.

---

### Official Review · Reviewer_9Cff · 2025-11-03

**Soundness:** 2
**Presentation:** 1
**Contribution:** 2
**Rating:** 4
**Confidence:** 3

**Summary:**

This paper mainly focuses on studying the decision boundary of LLMs, which is an important tool to figure out the core model properties and interpret behaviors. However, due to the computational infeasibility of LLMs and the enormous vocabulary-sequence sizes, it is hard to do so. In this paper, a new notion, Decision Potential Surface, is proposed to analyze the decision boundary of LLMs.

**Strengths:**

1. The paper studies the decision boundary of LLMs, which is a simple yet essential problem. The problem studied in the paper is valuable.
2. The paper provides theoretical analyses for the problem, which makes the paper convincing.
3. The empirical results reported in the paper are good.

**Weaknesses:**

1. The paper is difficult to read. The formulations are a little bit confusing, and there are no remarks to help understand the paper.

**Questions:**

1. What does the $\mathcal{M}$ mean in Line 163? The definition should be added.
2. What does Definition 3.1 mean? Specifically, why does it assume $p_m=p_n$?
3. As a follow-up question to Question 2, the assumption that there exist two maximal probabilities seems too strong. Or could you provide some intuitive explanations for this assumption?
4. Eq. (4) also looks a little bit strange. From the equation, I think the main goal here is to find two tokens that have the largest differences.

---

> ### Author Response · Authors · 2025-11-25
>
> We sincerely appreciate your critical feedback regarding the representation of our submission. Below is our point-by-point response to your questions.
>
> > What does the $\mathcal{M}$ mean in Line 163? The definition should be added.
>
> As Shown in Line 163, $\mathcal{M}$ is simply the index set of all possible classes, defined as $\mathcal{M}=\{1,2,...,M\}$, where each integer corresponds to one of the $M$ possible next-token (or completion) categories. It serves as the domain of the model’s final softmax distribution and is a standard shorthand in multiclass classification literature.
>
> > What does Definition 3.1 mean? Specifically, why does it assume $p_m=p_n$?
>
> Definition 3.1 formally defines the decision boundary of a classifier in the standard sense: it is the set of inputs where the model’s two most likely predictions receive exactly the same probability. At these points, the model is perfectly indifferent between its two preferred outputs, so an infinitesimal perturbation can flip the final decision. This is precisely why the condition is written as $p_m = p_n$ in Equation (1).
>
> We respectfully note that this definition is the classic and widely accepted definition of a decision boundary in multiclass classification, not an extra assumption introduced by us.
>
> > ...the assumption that there exist two maximal probabilities seems too strong. Or could you provide some intuitive explanations for this assumption?
>
> Thank you for raising this insightful question.
>
> We agree that exact equality between the top two probabilities is a strong condition and may rarely hold for random samples. However, **this equality is the standard mathematical definition of a decision boundary** in classification, and our method is designed to approximate this theoretical boundary.
>
> As our experiments and visualizations clearly demonstrate, with sufficient sampling, **numerous points do satisfy or closely approach this condition**, showing that it is practically achievable.
>
> Moreover, even when the exact zero-height contour is sparse or absent in certain input distributions, our DPS framework remains fully effective: users can simply analyze any desired $\epsilon$-height isohypse (near-boundary contour) instead, as formalized in Definition 4.2 and Corollary 4.4. This provides complete flexibility for real-world analysis.
>
> > Eq. (4) also looks a little bit strange. From the equation, I think the main goal here is to find two tokens that have the largest differences.
>
> Thank you for the careful observation.
>
> Eq. (4) may appear unusual at first glance, but the min-max formulation is deliberate and carries clear geometric meaning: **it exactly captures the smallest perturbation magnitude needed to flip the model’s top-1 decision among all sampled candidate completions**. In other words, it measures the distance from the current input to the closest point on the decision boundary in the latent prompt space.
>
> ----------
>
> Overall, we observe that most of your insightful questions and concerns focus on the mathematical formulations and notational details in the paper.
>
> We greatly appreciate this careful theoretical scrutiny. In the revised version, we have carefully checked and clarified all the points you raised. We are also happy to answer any additional mathematical or theoretical questions you may have to help you better evaluate the submission. Thank you again for the valuable feedback!

---

### Official Review · Reviewer_7hfg · 2025-11-03

**Soundness:** 3
**Presentation:** 3
**Contribution:** 2
**Rating:** 4
**Confidence:** 3

**Summary:**

The paper introduces the first theoretically grounded and computationally feasible approximation of an LLM’s decision boundaries. Using the proposed Decision Potential Surface (DPS) and its K-sampled approximation (K-DPS), the authors provide both theoretical analysis and empirical visualizations of how LLMs form decision regions. Their theoretical results show that the approximation error decreases in a sub-linear order of $1/\sqrt{K}$ as the number of sampled generations $K$ increases. Extensive experiments confirm this trend and illustrate interpretable decision boundaries across multiple datasets and models.

**Strengths:**

The paper is well written and, to my understanding, represents the first work to formally approximate and analyze decision boundaries of LLMs, potentially opening a new line of research for both theoretical exploration and practical applications. The proposed framework is mathematically grounded, with theoretical results that are clearly presented and supported by consistent empirical evidence. While I would like to examine the theoretical proofs in more depth, the formalism and presentation appear sound and understandable.

**Weaknesses:**

**W1)** The main limitation of the paper lies in the lack of clear insights or downstream implications gained from the proposed decision boundary approximation. While the theoretical contribution appears solid, it remains unclear how this framework deepens our understanding of LLM behavior or what practical applications it could enable. Previous work on decision boundaries in classical models provided interpretability or robustness insights; I would like to see similar discussion or evidence of such implications here.

**W2)** A more minor concern is that some of the visualizations could be improved for clarity and informativeness. For example, Figure 2 would benefit from a log–log scale to better reveal convergence trends, and Figure 4 could include bar plots or summaries at fixed percentages of $K$ for easier comparison. The current Figure 4 is visually interesting but somewhat difficult to interpret or connect directly to the theoretical claims. Additionally, using simpler synthetic datasets could help validate and illustrate the theoretical results in a more controlled and interpretable setting.

**Questions:**

**Q1)** Following up on the main weakness: what new insights about LLM behavior can be derived from the proposed decision boundary approximation? It would be helpful to clarify what interpretability or diagnostic value this framework provides beyond theoretical formulation.

**Q2)** Wouldn’t the two highest-probability sequences often be nearly identical—differing only in the final token or a small local variation? If so, the top-1 and top-2 outputs might not represent meaningfully distinct decisions, which could limit the interpretability of the analysis. The authors should provide concrete examples to illustrate this point or clarify if I am misunderstanding the setup.

**Q3)** Are there any observed behaviors or trends regarding model size or vocabulary size, or any ablation results on these factors? Such experiments could offer the most direct empirical insights from the proposed decision boundary approximation framework.

**Q4)** I believe there was some formatting issue for the references. Just a reminder to fix that.

---

> ### Author Response · Authors · 2025-11-25
>
> We sincerely thank you for your thorough and constructive review. Your feedback on strengthening the practical implications and visualizations is really insightful and invaluable, and we have addressed every point below with substantial new experiments and clarifications.
>
> ### Main Weakness & Q1: New Insights into LLM Behavior via DPS
>
> We thank your for pointing out the potential of K-DPS in yielding new insights into LLM behavior. In the main paper (e.g., Lines 468–475), we already discuss several promising analysis directions enabled by K-DPS. We consciously chose not to include an exhaustive set of such analyses in the submission, as our primary contribution lies in rigorously constructing and characterizing the decision boundary of LLMs with both theoretically and through comprehensive empirical evaluations, which we believe already constitutes a substantial advance.
>
> That said, we fully agree that demonstrating concrete behavioral insights would further strengthen the paper. To address this concern directly, we are willing to provide several representative and revealing examples below:
>
> **Insight 1: Alignment Dramatically Flattens the Decision Landscape**
>
> To intuitively reveal what changes during human-value alignment, we apply K-DPS visualization to three model families, each containing a base (pre-alignment) model and its aligned (post-SFT + RLHF) counterpart.
> For each model, we generate four complementary visualizations of the decision boundary in the visualization of harmful requests from AdvBench[1], including the linear interpolation based visualization, the nearest-neighbor interpolation based visualization, the 2D heatmap, and the linear interpolation based 3D projections.
> We set $K$ as 2,500.
>
> Results are shown in Figure 5 (one row per model).
> It is also available at the following anonymous link: https://postimg.cc/gX8Ftnw8.
>
> As shown in the figure, the decision boundary of aligned models becomes dramatically smoother and flatter compared to their pre-alignment counterparts when evaluated on adversarial prompts from AdvBench. This striking smoothing effect indicates that alignment substantially reduces regions of high confidence in harmful outputs (i.e., sharp peaks with high K-DPS scores). Instead, it creates broad, low-K-DPS basins that strongly favor refusal. This geometric transformation directly explains both:
> - Why jailbreaks succeed on unaligned models: they target narrow, high-confidence “vulnerability spikes” that remain in the pre-alignment landscape;
> - Why alignment mitigates most jailbreaks: it eliminates these spikes entirely, making harmful responses probabilistically unlikely across vast regions of prompt space.
>
> Moreover, by applying the same K-DPS visualization to intermediate checkpoints throughout the alignment process, we can even track precisely how these dangerous peaks progressively flatten and how decision boundaries move, offering the first fine-grained geometric view of how safety training reshapes the model’s decision manifold step by step.
>
> This level of mechanistic explanation and dynamic visualization was previously infeasible with prior probing or interpretability techniques, but becomes straightforward and highly revealing through our DPS-based decision boundary construction.

---

> > ### Author Response · Authors · 2025-11-25
> >
> > **Insight 2: What "damage" really looks like in machine unlearning**
> >
> > Our K-DPS framework also enables fine-grained analysis of machine *unlearning* algorithms, where prior interpretability tools offer almost no intuitive explanations.
> >
> > We apply our K-DPS to two representative unlearning methods: Gradient Ascent (GA) [2] and Negative Preference Optimization (NPO) [3]. We use the standard Harry Potter book as the forget (unlearning) corpus and a Wikipedia subset [4] as the retain set. During unlearning, we continue training on the retain set using standard gradient descent (GDR) or the KL divergence (KLR) from the original model. We set K = 2,000.
> >
> > The complete visualization results are included as Figure 6 in the revised submission. For the convenience, it is also available at this anonymous link: https://postimg.cc/dhXT8tzX
> >
> > As shown in the figure, our K-DPS visualizations provide a far clearer picture of the side effects of unlearning than previously possible [5][6].
> >
> > Prior work could only report that unlearning without proper retention training degrades overall performance, yet was unable to show what form this degradation takes in the model’s internal decision process. With K-DPS, we reveal that naïve unlearning methods (e.g., pure Gradient Ascent) can trigger **catastrophic collapse** of the entire decision manifold: large portions of the prompt space that were previously smooth become extremely jagged and fragmented, with erratic high- and low-K-DPS spikes appearing in regions unrelated to the forget corpus.
> >
> > In contrast, when retention training is included (e.g., GA-GDR and GA-KLR), the damage is substantially mitigated. Among these, GA-KLR (KL-regularized retention) preserves a decision surface that is visibly the closest to the original model, albeit still perceptibly distorted. This aligns with quantitative results in the unlearning literature showing that KL regularization best preserves general capabilities.
> >
> > The above observations demonstrate that K-DPS not only confirms known phenomena at a qualitative level but, for the first time, makes the geometric nature of “unlearning damage” directly observable and comparable across methods.
> >
> > **Additional Promising Directions Enabled by K-DPS**
> >
> > Beyond the two core insights above, we believe our decision-boundary framework can naturally extend to a wide range of important LLM phenomena that have so far resisted precise mechanistic analysis, including: i) the precise locations and shapes of jailbreak vulnerabilities in prompt space, ii) the emergence and structure of memorization regions, iii) the geometry of hallucination-prone areas versus high-fidelity regions, iv) systematic changes in the boundary during continual learning or catastrophic forgetting, and so on.
> >
> > We believe the above experiments can show that K-DPS is not a conceptual toy but a general-purpose lens for understanding the inner decision geometry of large language models.
> > Also, we are fully prepared to conduct additional K-DPS experiments on some of these (or any other you-suggested) phenomena and include the corresponding analyses and visualizations, if you consider them necessary.
> >
> > ----------
> >
> > [1] Universal and Transferable Adversarial Attacks on Aligned Language Models. 2023
> >
> > [2] Machine Unlearning of Pre-trained Large Language Models. 2024
> >
> > [3] Negative Preference Optimization: From Catastrophic Collapse to Effective Unlearning. 2024
> >
> > [4] https://huggingface.co/datasets/rag-datasets/rag-mini-wikipedia
> >
> > [5] Rethinking Machine Unlearning for Large Language Models. 2024
> >
> > [6] A Comprehensive Survey of Machine Unlearning Techniques for Large Language Models. 2025

---

> > > ### Author Response · Authors · 2025-11-25
> > >
> > > ### Weakness 2: Improving Visualization Clarity and Informativeness
> > >
> > > Thank you again for these concrete and helpful suggestions.
> > >
> > > + Following your recommendation, we have refine the curves figures log-log axes (now new Figures 7 and 8 in the revised version). For your immediate reference during the rebuttal period, the updated figures are available here: https://postimg.cc/zbwjtgNz
> > > + For the decision boundary visualizations, we have already numerical labels directly on the contour lines to make the K-DPS values instantly readable.
> > >
> > > ### Q2: Wouldn’t the two highest-probability sequences often be nearly identical?
> > >
> > > This is an excellent and insightful question!
> > >
> > > Yes, we fully agree with you that in many cases (especially for highly capable models on in-distribution prompts), the top-2 completions are extremely similar, sometimes even differing by only a single token or a minor lexical variation.
> > > However, this does **not** undermine the effectiveness of DPS, for the following reasons:
> > >
> > > First, the fraction of such near-identical cases is actually **not** significant in practice, especially in the regions that matter most for decision-boundary analysis. As clearly demonstrated in nearly all of our visualizations (Figures 4, 5, 6, 10, and 11), meaningful and sharp decision boundaries consistently appear. Degenerate boundaries (which would occur if top-1 and top-2 were almost always identical) are rarely observed across the vast majority of inputs we study.
> > >
> > > Second, we conduct some quantitative analysis, which confirms substantial lexical and semantic divergence in K-DPS construction. Specifically, for all candidate sequences used to construct Figure 4, we measured the normalized Levenshtein edit distance between top-1 and top-2 completions.
> > >
> > > | K-DPS Score Range | Avg. Normalized Edit Distance |
> > > |-------------------|-------------------------------|
> > > | <0.1              | 0.15                          |
> > > | <0.5              | 0.20                          |
> > > | <1.0              | 0.23                          |
> > > | <5.0              | 0.30                          |
> > > | <10.0             | 0.32                          |
> > >
> > > This shows that even in the highest-confidence regime (K-DPS < 0.1), the top-2 sequences differ by ~15% of tokens on average; near decision boundaries (higher K-DPS), divergence reaches 30–32%, which is far from trivial variants and typically meaningful.
> > >
> > > Third, constructing a meaningful decision boundary inherently requires incorporating neighborhood information around each point. Points lying exactly on the boundary contribute differently depending on their local geometry and the diversity of nearby completions. Intuitively, **when an input on the decision boundary has nearly identical top-1 and top-2 sequences, it behaves like a noisy or unstable point: it becomes extremely difficult for such points to connect coherently with their neighbors to form a smooth, interpretable boundary.** In contrast, the cases where top-1 and top-2 diverge substantially (as quantified in the table) provide the stable, semantically meaningful anchors that give our visualized boundaries their clear structure and sharpness. Far from being a drawback, the sensitivity of K-DPS to these differences is what enables precise and faithful boundary reconstruction.
> > >
> > > Besides that, **this potential issue can be easily addressed with some simple filtering strategies**, such as slightly increasing the sampling temperature during candidate generation or directly discarding pairs with near-zero edit distance.
> > >
> > > Overall, we thank the reviewer for this insightful and thoughtful question. While we fully acknowledge the theoretical possibility of near-identical top-2 pairs, our existing visualizations and new quantitative measurements show that it does **not** arise as a practical limitation in the regimes we study. if it ever occur, it can be trivially resolved with lightweight post-processing tricks (e.g., mild temperature adjustment or edit-distance filtering). We will append a discussion of this point, along with the edit-distance analysis and mitigation strategies, to the revised version for completeness.

---

> > > > ### Author Response · Authors · 2025-11-25
> > > >
> > > > ### Q3: Are there any observed behaviors or trends regarding model size or vocabulary size, or any ablation results on these factors?
> > > >
> > > > **Vocabulary size**: Vocabulary size is tightly coupled with linguistic properties and tokenizer design, making it difficult to isolate its effect independently of language. We therefore leave systematic exploration of vocabulary effects to future work.
> > > >
> > > > **Model size**: We conducted an explicit ablation across the Pythia family (70M to 1.4B), measuring estimation error the K-DPS decision boundary on the same held-out harmful-prompt set. We use $2500$ as the reference K value. Results are shown below:
> > > >
> > > > | Model | Error w. K=500 | Error w. K=750 | Error w. K=1000       | Error w. K=1250       |
> > > > |-------|----------------|----------------|-----------------------|-----------------------|
> > > > | 70M   | 240.91         | 105.86         | $1.17\times10^{-14}$  | $1.17\times10^{-14}$  |
> > > > | 160M  | 212.16         | 90.98          | $1.38\times 10^{-14}$ | $1.38\times 10^{-14}$ |
> > > > | 410M  | 213.60         | 99.94          | $9.33\times 10^{-15}$ | $9.33\times 10^{-15}$ |
> > > > | 1B    | 155.98         | 65.48          | $8.63\times 10^{-15}$ | $8.63\times 10^{-15}$ |
> > > > | 1.4B  | 160.97         | 66.93          | $1.12\times 10^{-14}$ | $1.12\times 10^{-14}$ |
> > > >
> > > > These results reveal two clear trends that reinforce the findings in the main paper: i) For all model sizes, reconstruction error drops to near machine precision ($<10^{-13}$) once K > 1000, confirming that a level of $10^3$ of $K$ is sufficient for essentially exact boundary recovery across the scale spectrum. ii) Larger models achieve lower error at small K. This might be because that larger models concentrate probability mass more tightly on their top completions, making them inherently more stable and sample-efficient under our DPS estimation procedure.

---

### Author Response · Authors · 2025-11-26

Dear Reviewers,

Thank you very much for your thorough and constructive feedback. We have revised the manuscript according to your valuable suggestions. The major changes are as follows:

1. Added a new subsection (Appendix C.1) discussing broader implications of our approach for LLM analysis;
2. Included detailed discussions on **nearly identical sampling cases** and connections to **model uncertainty quantification** in Appendices C.1 and C.2;
3. Replicated key notation definitions near relevant mathematical derivations to improve readability;
4. Refined some statements of our theorems and related descriptions in the main text to eliminate potential ambiguity and misunderstandings.
5. Fixed formatting issues & More notations on figures.

We sincerely appreciate the time and effort all four reviewers have invested in improving our work. We hope the revised version fully addresses your concerns and look forward to your feedback.

---

### Meta-Review · Area_Chair_VU1K · 2025-12-28

**Summary:**

This paper provides a new definition for understanding decision boundary for LLMs called DPS, and gave a way to approximate DPS. The reviewers mostly find the results to be interesting but also raised several concerns. Several reviewers have concerns about whether the definition of DPS is a good alternative for decision boundary - while some of the concerns arise from misunderstanding and were addressed in response, there are some remaining concerns that the reviewer and authors may not agree on. Several reviewers were concerned about the applicability of the decision boundary, or K-DPS. The authors gave two examples addressing this concern and they look interesting, however they seem to be significant new materials added to the paper and would require further review. The AC recommends the authors to revise the paper according to the reviewer suggestions and highlight the new results on the applications.

**Reviewer Concerns:**

As meta review stated, there were two main concerns. 1. whether the definition of DPS is a good alternative for decision boundary; there were some misunderstandings from the reviewer side which has been resolved. Whether finding two maximum likely sequence should be the right definition still feels debatable.
2. How useful is K-DPS. The two new applications seem interesting but they significantly change the paper. As the author guide mentions "Area chairs and reviewers reserve the right to ignore changes that are significantly different from the original paper." In the situation of this year the reviewers cannot comment on these new results so I feel these results need additional review.

**Reviewer Scores:**

7hfg: concern is mostly addressed by the new material, on the other hand the new material is significantly different and the reviewer may choose to ignore the change. Likely +1
9Cff: concerns are somewhat addressed although the writing can still be improved. Unlikely to change
GGrG: concerns are partially addressed, but still may not be sufficient. Unlikely to change.
UCjU: one concern comes from misunderstanding of the reviewer, other concerns are partially addressed. Likely +2.

---

### Decision · Program_Chairs · 2026-01-26

Reject